# Biased expectations about future choice options predict sequential economic decisions
Didrika S. van de Wouw ⓘ , Ryan T. McKay ⓘ & Nicholas Furl ⓘ ✉

Considerable research has shown that people make biased decisions in "optimal stopping problems", where options are encountered sequentially, and there is no opportunity to recall rejected options or to know upcoming options in advance (e.g. when flat hunting or choosing a spouse). Here, we used computational modelling to identify the mechanisms that best explain decision bias in the context of an especially realistic version of this problem: the full-information problem. We eliminated a number of factors as potential instigators of bias. Then, we examined sequence length and payoff scheme: two manipulations where an optimality model recommends adjusting the sampling rate. Here, participants were more reluctant to increase their sampling rates when it was optimal to do so, leading to increased undersampling bias. Our comparison of several computational models of bias demonstrates that many participants maintain these relatively low sampling rates because of suboptimally pessimistic expectations about the quality of future options (i.e. a mis-specified prior distribution). These results support a new theory about how humans solve full information problems. Understanding the causes of decision error could enhance how we conduct real world sequential searches for options, for example how online shopping or dating applications present options to users.

Often in everyday life, decisions must be made regarding options presented in sequence. For such scenarios we can ask ourselves, when should we stop evaluating new information and commit to a decision? This common real-life dilemma can be defined as an optimal stopping problem. For example, if one encounters a limited-time offer whilst shopping, should one accept it when it is available or pass on it and wait for a better one? If a doctor needs a healthy organ for transplant, should they use what is available now or risk waiting for a healthier one? If an animal welfare charity is visiting homes to find a suitable environment to rehome an animal, should they accept the currently visited home or continue to visit homes in hope of a better one? Problems like these can be referred to as "fiancé(e) problems", by analogy to decisions about whether to reject a current suitor in the hope of meeting better prospects in the future. We shall see below that solving many of these problems optimally is computationally challenging and that participants (when compared to the optimal solution) can under some circumstances show systematic decision biases. Our aim here is to delineate the experimental contexts in which participants exhibit these biases and to fit theoretical models to participants' choices to identify the computational mechanisms that give rise to these biases.

There are many variations of optimal stopping problem and their various computational solutions have been discussed in the fields of mathematics[1], behavioural ecology[2,3], economic decision making[4–6], cognitive

science[7] and neuroscience[8]. The computational solutions considered for optimal stopping problems are closely related to probabilistic reasoning and explore/exploit foraging decisions[9] and other sequential tasks that involve prospective reward prediction[10,11]. The availability of optimal computational solutions to optimal stopping problems enables researchers to use them as "ideal observer models", which can identify when people make suboptimal decisions, including decisions that reveal systematic biases.

We focus in the present study on a bias that arises for *"full information problems"*. This version of optimal stopping problem arguably most closely resembles real-world decision problems. Imagine an agent is searching for a new flat in a competitive market. The agent can sample a limited number of options in sequence (e.g. twelve flats can be viewed, one at a time) and must decide, for each option, whether to stop sampling and choose that option, under the condition that rejected options cannot be returned to later (e.g. refused flats are then offered to others and so become unavailable). Flat hunters in full information problems directly know the value of each option (e.g. how nice the currently viewed flat is or how much it costs). Full information problems can incorporate flexible payoff schemes (e.g. an agent might feel rewarded only if they achieve the best possible flat or their subjective reward might depend on the relative quality of whatever flat is chosen). Full information problems may involve a "cost to sample". Each

Royal Holloway, University of London, Egham, UK. ✉e-mail: nicholas.furl@rhul.ac.uk

time a new flat is visited (i.e. a new option is sampled), our flat hunter may incur calculable costs such as time, money or effort, which may be subtracted from the final achieved reward value and so can limit how many options are sampled. Finally, full information problems allow agents to harness their prior belief about the probability distribution that is generating their decision options (i.e. the generating distribution). When flat hunting, consumers can use these *prior expectations* about the housing market to prospectively compute the probability that an even nicer flat might be sampled if the current one is refused.

Here, we used experimental methods and computational modelling to test a raft of hypotheses related to an "undersampling bias". When the sampling behaviour of ideal observers is compared to that of human participants, humans often sample fewer options than is optimal[4,8,12–14]. To date, this undersampling bias has mainly been demonstrated for optimal stopping problems cast in economic scenarios in which options are represented as numbers (e.g. prices). Here, we have adapted the economic task first reported by Costa and Averbeck[8]. In our version, participants attempt to choose high-ranking smart phone prices.

However, undersampling bias is by no means universal. For example, some recent studies have reported full information problems associated with oversampling rather than undersampling[15,16]. These studies employed several different experimental and modelling methods that might have ameliorated the undersampling bias. Herein, we systematically manipulated each of these methods. We also explored computational mechanisms that can account for participants' errors on this task. We created theoretical computational models, each with a free "bias" parameter that skews otherwise optimal performance. Then we fitted these models over two pilot studies and three main studies, anticipating convergent explanations of bias across different experimental conditions with task methods. These models (as described in detail in the Methods) embody bias respectively due to (a) heuristic decision making, (b) bias due to an intrinsic value of sampling itself and (c) bias due to a mis-specified belief about the prior option distribution.

## Methods
### Paradigm summary
First, we briefly describe the general features of all our paradigms. Methods particular to the two pilot studies and Studies 1, 2, and 3 will be described in the succeeding sections. All study protocols were approved by the Royal Holloway, University of London College Ethics board and informed consent was obtained from all human participants in compliance with these protocols. Only Study 3 was pre-registered. We did not collect sex or gender data.

We implemented full information optimal stopping problems in which participants attempted to choose a competitive mobile phone contract. Prices used as options in all studies reported herein were for flagship models by the top brands (e.g. iPhone, Samsung, Huawei), on an up to 5GB plan with unlimited texts and minutes. The 90 prices were actual prices (in GBP) of 2-year contracts offered by various UK retailers, as harvested from internet advertisements in the year before the first data collection. The use of these real-world prices was intended to maximise the likelihood that the distribution of option values used in our studies would approximate the "true" generating distribution of smartphone price options in the participants' local market and thereby also approximate any prior expectations participants derived from their experience with smartphone contract prices.

In some conditions (discussed in more detail below: Pilot full, Study 1 ratings, Study 1 full, Study 2 and Study 3), the paradigm began with a "phase 1" ratings task, in which participants gradually viewed the full distribution of prices that could appear as options later by rating every price for its "attractiveness" or subjective value. As described below, some of the models we will be using make decisions based on objective values/raw prices and other models make decisions based on subjective values of the prices. The subjective values are derived from the ratings measured during phase 1. In phase 1, participants also could learn the "generating" distribution of option values and thereby establish expectations about the probabilities with which certain option values might appear in any given sequence, later in the optimal stopping task. The distribution of these ratings could then be used to

set the models' prior on its generating distribution of option values (See *Ideal observer optimality model* section).

Next, in the optimal stopping task, participants engaged with several fixed-length sequences of option values, populated by prices sampled randomly, without replacement, from the phase 1 generating distribution. In each sequence, participants sequentially encountered these prices and, for each price option, decided whether to reject that price (rendering it no longer accessible) and sample a new one or to take/choose that price. The decision to take a price terminated the search through the sequence and rendered all upcoming new prices no longer accessible. If the last price in a sequence was reached, that price became the participant's choice by default.

Our main behavioural dependent variable for the participants and all our models (the models are described below) was the number of samples before decision. A second performance measure of interest, the rank of the chosen prices, is reported in Supplementary Figs. 8–11 and 14. We computed frequentist and Bayesian t-tests using bf.ttest in the MATLAB bayesFactor toolbox https://github.com/klabhub/bayesFactor to compare these variables between participants and an Ideal Observer model and to compare participants' sampling rates between study conditions. We also used Bayesian t-tests to conduct comparisons between model performance measures (See section Theoretical models below). The Bayes factor toolbox follows Rouder et al.[17] by implementing a Jeffreys-Zellner-Siow prior, which involves a Cauchy prior on the effect size with a default scale factor of 0.71 = sqrt(2)/2, expecting medium effect size. The figures present results using this default scale factor, while all tables reporting Bayes factors include analyses of sensitivity to this Cauchy scale parameter. All frequentist and Bayesian tests are two-tailed. Dependent variables used in frequentist tests were checked for extreme deviations from assumptions using visual inspection of their distributions in conjunction with the Lilliefors and Brown–Forsythe tests. P-values for frequentist t-tests as well as the alpha values for their corresponding confidence intervals and assumption tests are always Bonferroni corrected for the number of pairwise comparisons per study.

### Methods specific to pilot studies
**Participants.** We recruited participants in both our pilot studies from the United Kingdom using the online data collection platform Prolific[18]. We enrolled 47 participants into Pilot Baseline and 50 participants into Pilot Full. Pilot baseline data were collected in October 2019. Pilot full data were collected in December 2019.

**Procedures.** We used Gorilla Experiment Builder[19] to create and host Pilot baseline and Pilot full studies. For Pilot baseline, we attempted to replicate participant undersampling bias as shown in previous studies[8,12], in which participants sampled fewer options than the same Ideal Observer that we used herein. Therefore, we matched the methods of Pilot baseline to Costa and Averbeck[8] as closely as was practical, while adapting the paradigm for an online data collection setting. As Pilot baseline is representative of all our paradigms, we illustrate it in Fig. 1, while screenshots from other conditions are shown in Supplementary Figs. 1, 2 and 3. There was no phase 1 ratings task in Pilot baseline. In the Pilot baseline optimal stopping task, participants attempted to choose one of the top three ranked smartphone prices out of each option sequence. The option value screen also presented the previously rejected option values and the number of options remaining in the sequence. Each sequence used a fixed order of 12 option values, so a given sequence's option values and their order within the sequence was identical for every participant (and corresponding models), although the sequences themselves were intermixed randomly.

Like Costa and Averbeck[8], we rewarded participants financially for choosing one of the top three options in the sequence. Participants in Pilot baseline earned £0.12 per sequence if they chose the best price in the sequence, £0.08 if they chose the second-best price, £0.04 if they chose the third-best price, and £0 if they chose any other option. These performance-based bonus payments were earned on top of a flat fee, which for all our studies was set in line with Prolific's recommended pay of £7.50 per hour

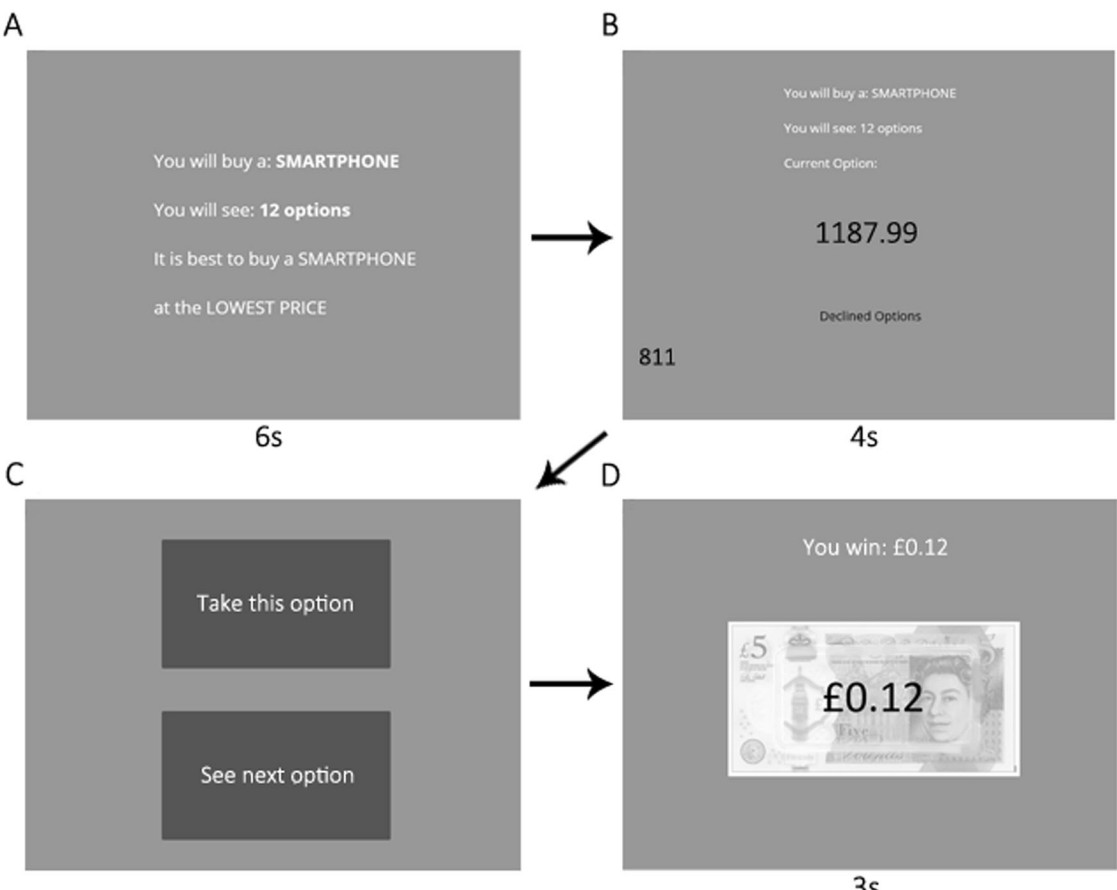

**Fig. 1 | Baseline pilot paradigm.** Participants are instructed to buy a smartphone (**A**), then they view option values for a fixed duration (**B**), choose to take each option or sample another (**C**) then, upon taking an option, receive feedback (**D**) about the reward value of their choice (monetary remuneration when top three ranked options are chosen).

(participants typically finished the study in considerably less time than an hour). Once a choice was made, participants viewed a feedback screen that informed them of their winnings for that sequence. The paradigm implemented fixed screen timings, meaning that participants automatically advanced through the screens, except when asked to decide ('Take this option' or 'See next option'), in which the paradigm waiting for the response. Participants were warned about the timing in the instructions preceding the task.

For Pilot Full, we were interested in whether participant undersampling bias would continue to replicate using the same economic smartphone price task, but when implementing the "full" complement of methods particulars adapted from studies that revealed oversampling bias instead of undersampling bias[15,16]. The logic is that, if any of these methods features is responsible for the oversampling bias seen in these earlier reports, then Pilot full should produce an oversampling bias, which would contrast with the undersampling bias we expected to see in Pilot baseline.

Pilot full added an initial ratings phase (Supplementary Fig. 1, left), in which participants rated the "attractiveness" of the price, defined in the instructions as a willingness to purchase a phone at that price. Ratings were made by mouse click on a sliding scale from 1 to 100, with the slider only appearing after the first click - to avoid slider biases[20] - with the selected rating value shown above the slider. Participants rated 180 prices, presented one at a time in a random order, and comprising the 90 unique prices, each rated twice. The average over the two ratings for each price was then used as the subjective value input to the SV versions of the models. In Pilot full, the mean (over participants) Pearson's correlation coefficient between the two ratings was 0.83. A blue progress bar was shown continuously at the bottom of the screen to visualise participants' progression through the ratings phase.

The optimal stopping (second) phase of Pilot full (Supplementary Fig. 1, right) included five sequences of 12 option values each. As in Pilot baseline, the option values in each sequence were fixed in advance but the sequences' order was randomised. Unlike Pilot baseline, once participants chose one of the options, they then had to advance by button press through a series of grey squares that replaced the remaining options in that sequence. This was intended to discourage participants from finishing the study early by choosing earlier options. Also unlike Pilot baseline, the optimal stopping task was entirely self-paced - participants advanced by using their mouse to click on the buttons on the screen. After finishing a sequence, participants were directed to a feedback screen displaying their chosen price and the text: "This is the price of your contract! How rewarding is your choice?". Participants responded to this question using a slider scale ranging from not rewarding (1) to very rewarding (100). The purpose of this rating activity was only to provide feedback to the participants about the quality of their choices, in lieu of the bonus payoff screen in Pilot baseline, and to encourage participants to reflect upon the choice's reward value before moving on to the next sequence. These ratings do not provide hypothesis-relevant data and were not analysed. Participants were reimbursed a flat fee only - no bonus monetary payoff was awarded.

### Methods specific to Study 1
**Participants.** As in the pilot studies, participants in Study 1 were enrolled from Prolific's pre-screening facility to ensure that all participants were residents of the United Kingdom, to maximise familiarity with current UK smartphone market prices, denominated in GPB. We enrolled independent participant samples into each of six conditions (See Procedures), targeting fifty participants in each condition (chosen based on

**Table 1 | Summary of conditions for Study 1**

| | | Study 1 condition name | | | | | |
|---|---|---|---|---|---|---|---|
| | | **Baseline** | **Full** | **Squares** | **Payoff** | **Timing** | **Ratings** |
| Task feature | Grey squares | | × | × | | | |
| | No monetary payoff | | × | | × | | |
| | Self-paced timing | | × | | | × | |
| | Rating phase | | × | | | | × |

our pilot studies, whose sample sizes proved sufficient to discriminate participant and Ideal Observer sampling rates). However, because of a technical difficulty with the participant recruitment platform, we overshot our data collection target by two participants, one in the payoff condition and one in the ratings condition. Data were collected in February 2021.

**Procedures.** The study was developed using the experiment hosting software Gorilla Experiment Builder[19]. We implemented six conditions in Study 1, which systematically manipulated the presence or absence of four key task features. These task features were those that varied between Pilot baseline and Pilot full and were manipulated to test if any were capable of modulating participants' or Ideal Observers' sampling rates. These task features are summarised in the rows of Table 1. Next, we will cover each condition in turn.

The *baseline condition* (Fig. 1) was nearly identical with the Pilot baseline study, except that it implemented seven sequences instead of five. That means that Study 1 baseline and Pilot baseline both adapted methods from studies showing undersampling[8,12]. Study 1 baseline is "baseline" in the sense that this condition possesses none of the additional methodological features associated with oversampling[15,16], and it serves as the basis for comparison against the other conditions, which each add one or more of these additional methodological features. Like Pilot baseline, we fixed in advance the option values and their order within each of the sequences and then presented these fixed-option sequences in random order. However, in this case, to avoid as homogenous a set of sequences as was used in Pilot baseline, we created 10 such fixed sets of sequences and each participant was randomly assigned to one of these sets. This procedure was also implemented in all the conditions that were based on Study 1 baseline, which we described below (i.e. ratings, payoff, squares, timing). The *full condition* was identical to the Pilot full study (Supplementary Fig. 1), except that it used seven sequences instead of five. The mean (over participants) Pearson's correlation coefficient between the two ratings for each price collected in the first phase was 0.87. The *ratings condition* was the same as the baseline condition with the exception that it added the same initial rating phase as Pilot and Study 1 full conditions (Supplementary Fig. 1, left), but still used the same optimal stopping task as the baseline condition (Fig. 1). In this condition, the correlation between the two ratings for each price (averaged over participants) was 0.81. The *payoff condition* (Supplementary Fig. 2) was the same as the baseline condition with the exception that participants did not receive the monetary incentivisation that they did in the baseline condition. Instead, participants were instructed to make choices to maximise the number of stars. Then, instead of receiving feedback regarding their earned bonus payments on the feedback screen (as in the baseline condition), participants were shown pictures of the number of stars that they earned for their choice: either five stars, three stars or one star, if they chose respectively the best, second best, or third best price in the sequence. The *squares condition* (Supplementary Fig. 3) was the same as the baseline condition with the exception that, once participants had chosen an option that was not the last option, they had to press a key to advance through grey squares that replaced each forgone option until the end of the option sequence. The *timing condition* was the same as the baseline condition with the exception that this condition incorporated a "next" button in the top right corner of every option screen. This button ensured that participants

controlled the pace of the study, rather than screens advancing automatically with fixed timings.

## Methods specific to Study 2
**Participants.** One hundred fifty-one participants based in the UK enrolled, using the participant recruitment platform Prolific. Data were collected in March 2023.

**Procedures.** Study 2 once again implemented the full condition, but now employing an enhanced design that fully randomised all option values and dramatically increased the sample size. This enabled us to conclusively resolve some discrepancies between the results of Pilot full and Study 1 full (See Results section for more information) and to provide a statistically more powerful comparison of models. The study was developed in Javascript jsPsych 7.3.1[21]. In phase 1, participants rated two-times each the same 90 smartphone prices used in our studies above, with all stimuli presented in one random sequence. Prices appeared above a 1 to 100 scale, and participants indicated the "attractiveness" of each price via mouse click on the scale. The mean (over participants) Pearson's correlation coefficient between the two ratings for each price was .85. Next, participants performed an optimal stopping task with six sequences of 12 price option values, randomly sampled without replacement from the 90 prices. The study implemented participant-paced screen timing. There were no grey squares. Instead, upon choice, the paradigm proceeded directly to the feedback screen. The feedback screen appeared as described above for Pilot full and Study 1 full. Participants were instructed to choose the best possible price.

## Methods specific to Study 3
Study 3 once again implemented the full condition, but this time manipulating sequence length. As participants in our preceding studies in this did not change their sampling rates to a statistically detectable degree, our goal for Study 3 was to test whether participants would do so at all. Costa and Averbeck[8] previously showed that sequence length both increased participants' sampling rates and increased the size of their undersampling bias, compared to the Ideal Observer, and we attempted to replicate those findings here.

Study 3 was preregistered at https://osf.io/vcf7u in April 2023, with the data collected shortly thereafter. We enrolled 140 participants from the UK using Prolific. Fifty percent random assignment of participants to each group yielded 65 participants with 14 options and 75 participants with 10 options (which is a slight deviation from our pre-registered plan of 70 participants per group). As explained in the pre-registration, the sample size was intended to double that of Costa & Averbeck[8] (who used a more powerful repeated-measures design and who were able to use more trials per participant in-lab, while we needed a shorter online study). The procedures were identical to Study 2, using the same jsPsych code, merely changing the sequence length of the optimal stopping phase of the study. The averages (over participants) of the Pearson's *r* values computed between the two phase 1 ratings to each price were 0.88 for the 10 option condition and .84 for the 14 option condition.

**Ideal observer optimality model.** To analyse the optimal stopping task data in all these studies, we compared the number of options our participants sampled before choosing an option to that of the Ideal Observer.

The Ideal Observer is a benchmark of optimality, for which performance is Bayes-optimal. This finite-horizon, discrete-time, Markov decision process model has been used in previous studies[8,12,15,16]. The Bayesian version of the optimality model for the full information problem builds on the classic Gilbert and Mosteller model[22]. Models try to predict upcoming option values, with these expectations derived from the model's belief about the distribution from which future options are assumed to be generated (i.e. the generating distribution). More precisely, the utility $u$ for the state $s$ at sample $t$ is the maximal action value $Q$, out of the available actions $a$ in $A$. These action values in turn depend on the reward values $r$ and the probabilities of outcomes $j$ of subsequent states (i.e. the generating distribution), weighted by their utilities.

$$u_t(s_t) = \max_{a \in A_{s_t}} \{r_t(s_t, a) + \int_s p_t(j|s_t, a)u_{t+1}(j)d_j\} \quad (1)$$

The terms appearing inside the curly brackets are taken collectively as the action value $Q$. $r_t(s_t, a)$ is the reward that would be obtained in state $s$ at sample $t$ if action $a$ is taken. The model described here reduces $r$ by costs incurred by sampling again using a "cost to sample" penalty term $C$. See formula for $r_t(s_t, a = sample\ again)$ below. As there was no extrinsic cost-to-sample in any of our experimental designs herein, $C$ was always fixed to zero for the Ideal Observer. The integral is taken over the possible states after the current sample. Each of these states is weighted by the probability of transitioning into it from the current state, given by $p_t(j|s_t, a)$, as derived from the generating distribution.

The utilities for sampling again are computed based on backwards induction (See Supplementary Methods: detail on backwards induction). The model first considers the utility for the final sample $N$ in the sequence, which is simply the reward value associated with the $N$th state (because taking the option is the only available action for the final sample in a sequence).

$$u_N(s_N) = r(s_N)\ for\ all\ s_N \in N \quad (2)$$

Next, the model works backwards through the sequence, iteratively using the aforementioned formula for $u_t(s_t)$ when computing each respective action value $Q$ for taking the option and declining the option for each $t$. Whenever the reward value of taking the current option is considered, the reward function $R$ assigns reward values to options based on their ranks. $h$ represents the relative rank of the current option.

$$r_t(s_t, a = take) = \sum_{i=1}^{N} p(rank = i) * R(i + (h - 1)) \quad (3)$$

In contrast, the reward value of sampling again is simply the cost to sample $C$.

$$r_t(s_t, a = sample\ again) = C \quad (4)$$

This customisable $R$ function allowed us to examine how the Ideal Observer changes its sampling strategy under the different reward payoff schemes used in our studies. Pilot full, Study 1 full, Study 2 and both conditions in Study 3 all involved instructing participants to try to choose the best price possible. In study conditions using these instructions, we implemented a continuous payoff function (resembling that of the classic Gilbert & Mosteller formulation), in which the relative rank of each choice would be rewarded commensurate with the value of its associated option. In Pilot baseline and the baseline, squares, timing, and prior conditions of Study 1, we adapted the payoff scheme to match participants' instructions that they would be paid £0.12 for the best rank, £0.08 for the second-best rank, £0.04 for the third best rank and £0 for any other ranks. Lastly, in the payoff condition of Study 1, we programmed the reward payoff function to match participants' reward of 5 stars for the best rank, 3 stars for the second-best rank, one star for the third-best rank and zero stars for any other ranks.

Another feature added to our implementation of the Ideal Observer, compared to the Gilbert & Mosteller base model, is the ability to update the model's generating distribution from its experience with new samples in a Bayesian fashion, instead of this generating distribution being specified in advance and then fixed throughout the paradigm. This Bayesian version of the optimality model treats option values as samples from a Gaussian distribution with a normal-inverse-$\chi 2$ prior. The prior distribution is initialised before experiencing any options with four parameters: the prior mean $\mu_0$, the degrees of freedom of the prior mean $\kappa$, the prior variance $\sigma^2_0$ and the degrees of freedom of the prior variance $v$. The $\mu_0$ and $\sigma^2_0$ parameters of this prior distribution are then updated by the model following presentation of each newly sampled option value as each sequence progresses.

Here, we set the prior values of $\mu$ and $\sigma^2$ in two possible ways: objective value and subjective values versions. In some previous studies of optimal stopping for price decisions[4], the mean and variance of the generating distribution has been fixed in advance by the mean and variance of the distribution of objective prices. We implemented an objective values version of the Ideal Observer in this way for all the study conditions reported herein. This objective values procedure for the Ideal Observer assumes that the raw prices can be treated as a proxy for participants' subjective value of the prices, so an Ideal Observer that optimises only the raw prices when making decisions would therefore be an appropriate basis for comparison with participants. However, we also had direct access to participants' subjective values of options in some conditions (Pilot full, Study 1 full condition, Study 1 ratings condition, Study 2 and both sequence length conditions of Study 3), due to the presence of the initial rating phase, and so we could also build a subjective values version of the Ideal Observer. This second way of computing the Ideal Observer assumes that participants' subjective valuation of prices may not necessarily exactly equal the raw price values, especially in their scaling, which may be relevant to full information problems. We used each participants' individualised ratings (subjective valuations) of the prices as option values input to the subjective values version of the Ideal Observer, and we used the mean and variance of individual participants' ratings distributions when initialising the prior of the generating distribution of the Ideal Observer.

Because conditions with an initial rating phase had objective and subjective values versions of the Ideal Observer, with each version providing separate optimality estimates, we were able to test the hypotheses that the use of objective or subjective values when modelling (a) affects the strategy taken by the optimality model and (b) changes the assessment of participant bias. We ensured for both objective values and subjective values versions of the models that better options were always more positively-valued such that the models were always solving a maximisation problem. We further ensured that estimated parameters for both objective values and subjective values versions of the models would be on the same scales by reflecting the objective prices around their mean. Then we rescaled those values to span 1 (the highest/worst price) to 100 (the best price). These reflected and rescaled objective values were then used in objective values models when computing the prior generating distribution, and when inputting price values to the model as option values. Subjective values were already rated by participants on this same 1 to 100 scale.

**Theoretical models.** The purpose of the Ideal Observer described above was to assess bias, not to theoretically explain participants' bias. By the definition of an ideal observer, its parameter values should be fixed to ground truths established by the experimental design. Because of this feature, however, optimality models, in general, are not appropriate for use as theoretical models of potentially biased human sampling and choice behaviour, without modification added to account for sources of individual variability in bias. That is, the Ideal Observer only models the computations leading to accurate choices but not to systematic sources of error. To better understand which computations might be responsible for participants' errors, we formulated several theoretical models and fitted them to participants' take option versus sample again choices. As

mentioned above with respect to ideal observer models, some previous studies have implemented models that aim to optimise the objective values of choices[4,7,8,12] while other model implementations optimise subjective values of those options, obtained via a separate rating task[15,16]. Because there is no obvious determination of which procedure is correct, we implemented both objective values and subjective values versions of all our theoretical models, whenever a study condition involved a preceding rating task that enabled both model implementations. Then, we could assess using model comparison whether both objective values and subjective values versions of the models best fit human participant choices, or whether these two model varieties are relatively interchangeable (which we in fact discovered, see Results).

For every sample, the probabilities of the two available choices (take current option versus sample again) were computed by transforming action values from each model to probabilities using Softmax and then summing negative log likelihoods over choices for each participant. In each model, we freed one theoretically interpretable key parameter (these free parameters and their models are described below) and the inverse temperature parameter beta from the Softmax function (the starting value for beta was always 1 and the fitting of beta was bounded between 0 and 100). During parameter recovery (Supplementary Results: Parameter recovery), we confirmed that varying each of the key theoretical parameters could indeed modulate the sampling rate throughout its entire range (Supplementary Fig. 5). The two free parameters per model were fitted using fminsearchBND.m in MATLAB (Mathworks, Natick MA). Parameter recovery analyses for three of the models we consider and describe below showed at least adequate correlations between configured and recovered parameters (Supplementary Fig. 4): The Cut Off heuristic and the Cost to Sample and Biased Prior models (We will describe these three models in detail next and they are summarised in Table 2). These three models also showed strong correlations between sampling rates associated with configured parameters and sampling rates associated with recovered parameters. Two other theoretically motivated models—the Biased Values and Biased Rewards models (See Supplementary Results: Parameter recovery)

—performed poorly during parameter recovery and so were excluded from the formal model comparison. We implemented two parallel model comparison methods based on negative log-likelihood values converted to Bayesian information criterion (BIC) values. For the first model comparison method, we submitted the BIC values to repeated measures pairwise statistical tests using Bayes factors to ascertain whether pairs of models differed or had equivalent BIC values on average over participants. Better fitting models show statistically lower BIC mean values. For the second model comparison method, we computed which model had the lowest (best) BIC for each participant and then plotted histograms to ascertain which model(s) dominated the others in terms of participant "wins". The model that best fitted the most participants presumably was the sampling strategy most often used by participants in our sample. We also confirmed the practical success of our model fitting procedures by fitting the three theoretical models to data simulated by the three models and found reasonably accurate model recovery performance (Supplementary Fig. 7).

We now turn to descriptions of the three theoretical models that survived parameter recovery and graduated to model comparison. The objective and subjective values versions of the *Cut Off heuristic* derive from the mathematically optimal solution to the "Secretary problem"[1,23], an optimal stopping problem with a mathematical solution that is relatively simple, as it is constrained by numerous required assumptions that need not hold for full-information problems. Namely, the secretary problem solution assumes the agent uses no prior knowledge of the generating distribution, considers only relative ranks of option values and feels rewarded only when choosing the top-ranked option. Although the Cut Off heuristic derives from the optimal solution to the secretary problem, which makes different assumptions than the optimality solution of the full information problem we consider here, Todd and Miller[24] propose that the Cut Off heuristic might nevertheless be robust to violations of the secretary problem assumptions and, being a heuristic, would be relatively simple for humans to compute on the fly in realistic settings. More specifically, Todd & Miller propose that such a Cut Off heuristic explains undersampling bias

## Table 2 | Key features of fitted theoretical models

| Model name | Free parameters | Strategy |
|---|---|---|
| Cut Off heuristic | Cut off, Softmax beta | Chooses no option until after a "cut off" number of samples, then chooses next option with highest relative rank. |
| Cost to Sample | Cost to sample, Softmax beta | Ideal Observer, but sampling can be perceived as costly. |
| Biased Values | Option value threshold, Softmax beta | Ideal Observer, but only option values above a threshold can be considered. |
| Biased Prior | Constant added to prior mean, Softmax beta | Ideal Observer, but expected value (mean) of prior option value distribution can be reduced by a constant. |

**Fig. 2 | Human participants' numbers of samples to decision for all studies.** Significant frequentist pairwise differences between condition means are shown as green horizontal lines ($p < 0.05$, Bonferroni corrected for the number of pairs in each study) for Study 1 full (n = 50 participants), Study 1 ratings (n = 51), baseline (n = 50), squares (n = 50), timing (n = 50), payoff (n = 51), Study 2 10 options (n = 75) and 14 options (n = 65). Only greater sampling for longer sequences in Study 3 proved significant. Null effects were concluded based on $BF_{01} > 3$ (i.e. at least moderate evidence for equal means). Such pairs are connected by magenta horizontal lines. Detail on pairwise statistical tests depicted here can be found in Table 3 and Supplementary Table 3. Boxplots reflect first, second (median) and third quartiles, while whiskers reflect 1.5 interquartile range. Points reflect individual participant mean values.

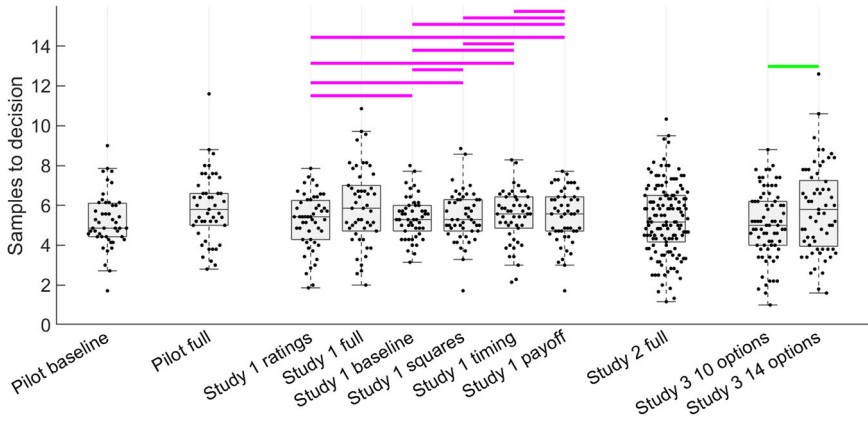

because the heuristic can perform nearly-optimally (on secretary problems) while incurring fewer samples, which "satisfices" under conditions where the problem involves a ground truth cost for new samples (note that the Cut Off heuristic has no formal cost to sample parameter). This heuristic has previously been fitted to human behaviour on full-information optimal stopping problems, although little evidence was found favouring it in that study[4]. The Cut Off heuristic chooses to sample again for every option until it reaches a cut-off sequence position, which is fitted as the key theoretical free parameter. Then, the model continues to sample until it reaches the next option with the highest relative rank so far. Here, we used the optimal cut-off value (37% of the sequence length, rounded to the nearest integer) as the starting value during model fitting and the

parameter search was bounded between 2 and the sequence length minus 1 (as the learning period defined by the cut-off must contain at least one sample and be followed by at least one sample available for choice). Cut-off values below the optimal value lead to undersampling and cut-off values above the optimal value lead to oversampling.

We also considered objective and subjective values versions of the *Cost to Sample model*. These use the Ideal Observer for the full information problem described above as a base, while also assuming that participants' otherwise rational Bayesian computations can be biased by a free parameter value. In the case of the Cost to Sample models, the fitted parameter to account for such bias was the cost to sample value $C$ (See computation of $r_t(s_t, a = sample\ again)$ in the Ideal Observer Optimality Model

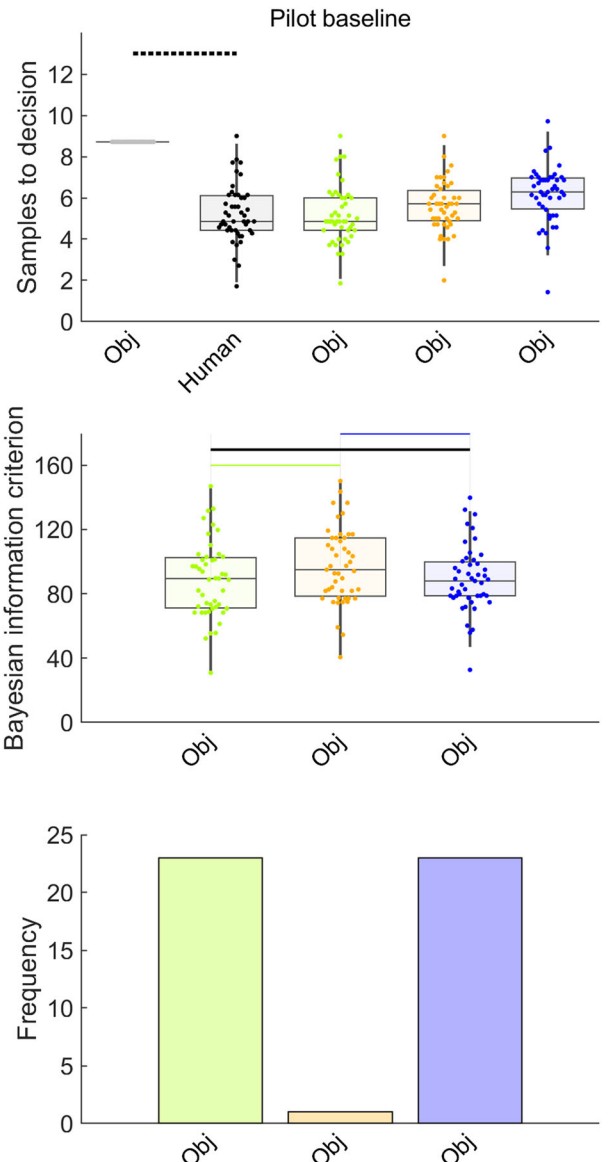
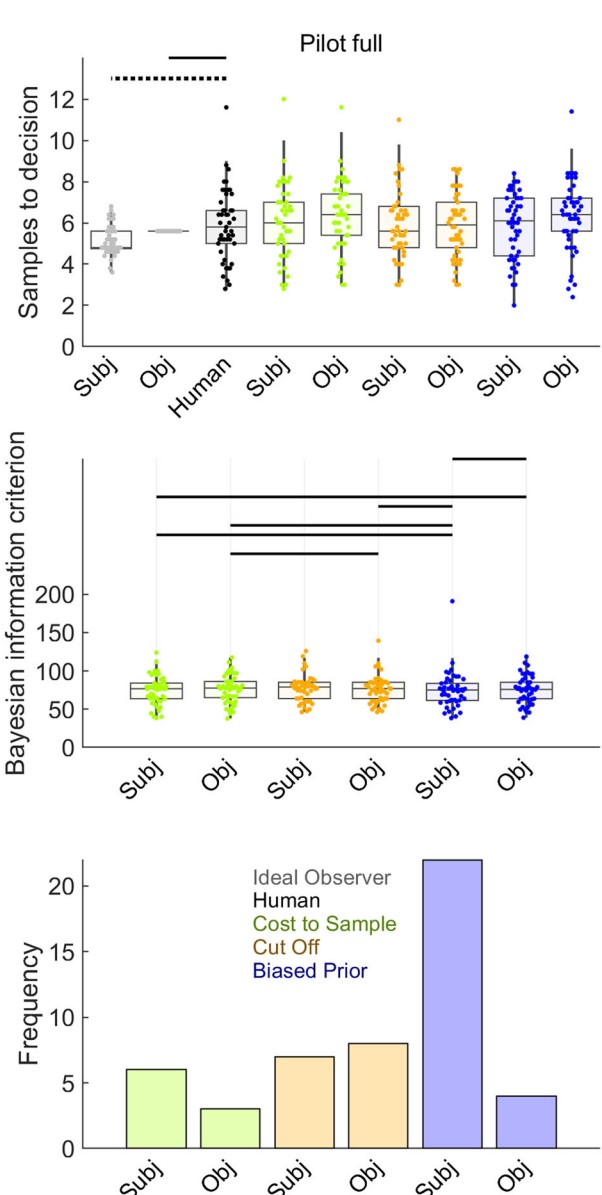

**Fig. 3 | Model comparison for pilot studies.** Results from Pilot baseline (n = 47 participants) are shown in left column, with Pilot full (n = 50) in the right column. Human participant data are reproduced from Fig. 2. In the first row, horizontal solid lines link samples data for human (black points) and Ideal Observer (grey points) when $BF_{01} > 3$ (at least moderate evidence for equal means) while dotted lines indicate when $BF_{10} > 3$ (at least moderate evidence for different means). More statistical details for these pairwise tests can be found in Supplementary Table 1. Pilot baseline participants undersampled, compared to the Ideal Observer, though participants in Pilot full did not. The second row shows BIC values, where lower values indicate better model fit. Horizontal lines are shown in the colour corresponding to

the point spread of the better-fitting model when $BF_{10} > 3$ or in black when $BF_{01} > 3$. Further statistical details for these pairwise Bayesian mean difference tests can be found in Supplementary Table 2. Though mean BIC did not favour one leading model in the pilot studies, the third row in the full condition suggests that Biased Prior model best fitted the most participants. Model point spread data colours (See also legend in lower right panel): Cost to Sample (green), Cut Off (orange), Biased Prior (blue). Boxplots reflect first, second (median) and third quartiles, while whiskers reflect 1.5 interquartile range. Point spreads reflect individual participant mean values. Abbreviations: Subj = Models that make choices about subjective values; Obj = Models that makes choices about objective values.

section above). In the Cost to Sample model, participants would undersample if they intrinsically perceive sampling as costly and so adopt a negatively valued $C$, whereas they would oversample if they perceive sampling as rewarding as so adopt a positive $C$. We initialised model fitting with a starting $C$ value of 0 (i.e. the optimal value) and, during fitting, bounded $C$ to be between –1 and 1 (The payoffs we used during model fitting were scaled to be between 0 and 1 and C values are specified on that scale).

We used a similar approach when building the subjective and objective values versions of the *Biased Prior model*. In this model, we added a new free parameter to $u_0$, the mean of the prior generating distribution. Negative values of this parameter can bias an agent to compute pessimistic estimates of future option values by shifting the prior mean (i.e. expectation) to be lower. This can lead to undersampling by making the current option appear more appealing compared to the artificially deflated expectation of option values resulting from continued sampling. We initialised model fitting with a starting value of 0 (i.e. the optimal value) and the biased prior parameter

was bounded during fitting to be between –100 and 100 (when compared to option values scaled between 0 and 100).

### Reporting summary

Further information on research design is available in the Nature Portfolio Reporting Summary linked to this article.

## Results
### Pilot Studies

As the two pilot studies are separate studies, with data collected at somewhat different times, we will descriptively, rather than statistically, compare them. Study1 will also implement baseline and full conditions like these, and so they will be statistically compared in that study. Recall from the Methods that Pilot baseline aimed to replicate studies showing undersampling[8,12] by adapting their methods. In contrast, Pilot full took the rather different methods used in studies showing oversampling[15,16] and adapted their

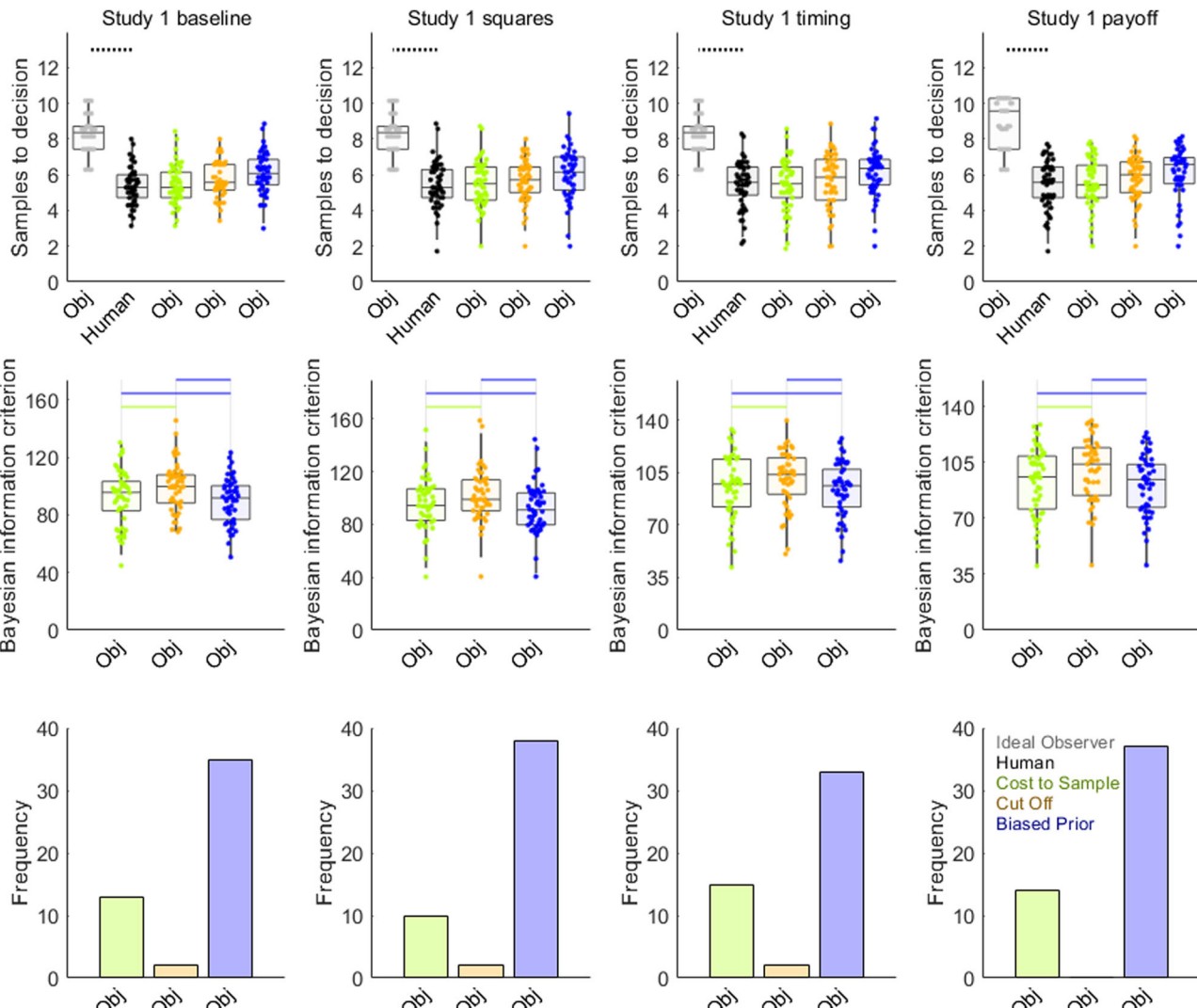

**Fig. 4 | Model comparison for Study 1 conditions with no first phase.** From left to right, columns show results from baseline ($n$ = 50 participants), squares ($n$ = 50), timing ($n$ = 50) and payoff ($n$ = 51) conditions. Human participant data are reproduced from Fig. 2. In the first row, horizontal solid lines link samples data for humans (black points) and Ideal Observer (grey points) when $BF_{01}$ > 3 (at least moderate evidence for equal means) while dotted lines indicate when $BF_{10}$ > 3 (at least moderate evidence for different means). Statistical details for these pairwise tests can be found in Supplementary Table 1. Every condition replicated robust participant undersampling. The second row shows BIC values, where lower values indicate better model fit. Horizontal lines are in the colour corresponding to the

point spread of the better-fitting model when $BF_{10}$ > 3 or shown in black when $BF_{01}$ > 3. Further statistical details for these pairwise mean difference tests can be found in Table 4. All four conditions replicated Biased Prior model (blue) as the best-fitting model. Similarly, the third row shows that the Biased Prior model best fitted the most participants. Model point spread data colours (See also legend in lower right panel): Cost to Sample (green), Cut Off (orange), Biased Prior (blue). Boxplots reflect first, second (median) and third quartiles, while whiskers reflect 1.5 inter-quartile range. Point spreads reflect individual participant mean values. Abbreviations: Subj = Models that make choices about subjective values; Obj = Models that makes choices about objective values.

methods to an economic context. We were interested in whether any of the methods used in Pilot full would lead to oversampling. Figure 2 shows the mean number of samples to decision made by human participants, which yielded similar numbers of samples for both pilot studies, with only a slight numerical increase for Pilot full.

Figure 3 (See also Supplementary Table 1) shows results from the comparison of human participants' sampling with sampling of the Ideal Observer

and theoretical models. As expected, we successfully replicated undersampling in the Pilot baseline condition (Fig. 3, upper left), where participants sampled fewer options than the Ideal Observer (Cohen's $d = -2.52$). All the theoretical models, after fitting to Pilot baseline data, resembled the participants to some degree, as they all showed some undersampling, compared to the objective values version of the Ideal Observer. Bayesian pairwise tests (Fig. 3, left column, second row and Supplementary Table 2), showed that the Cost to Sample and

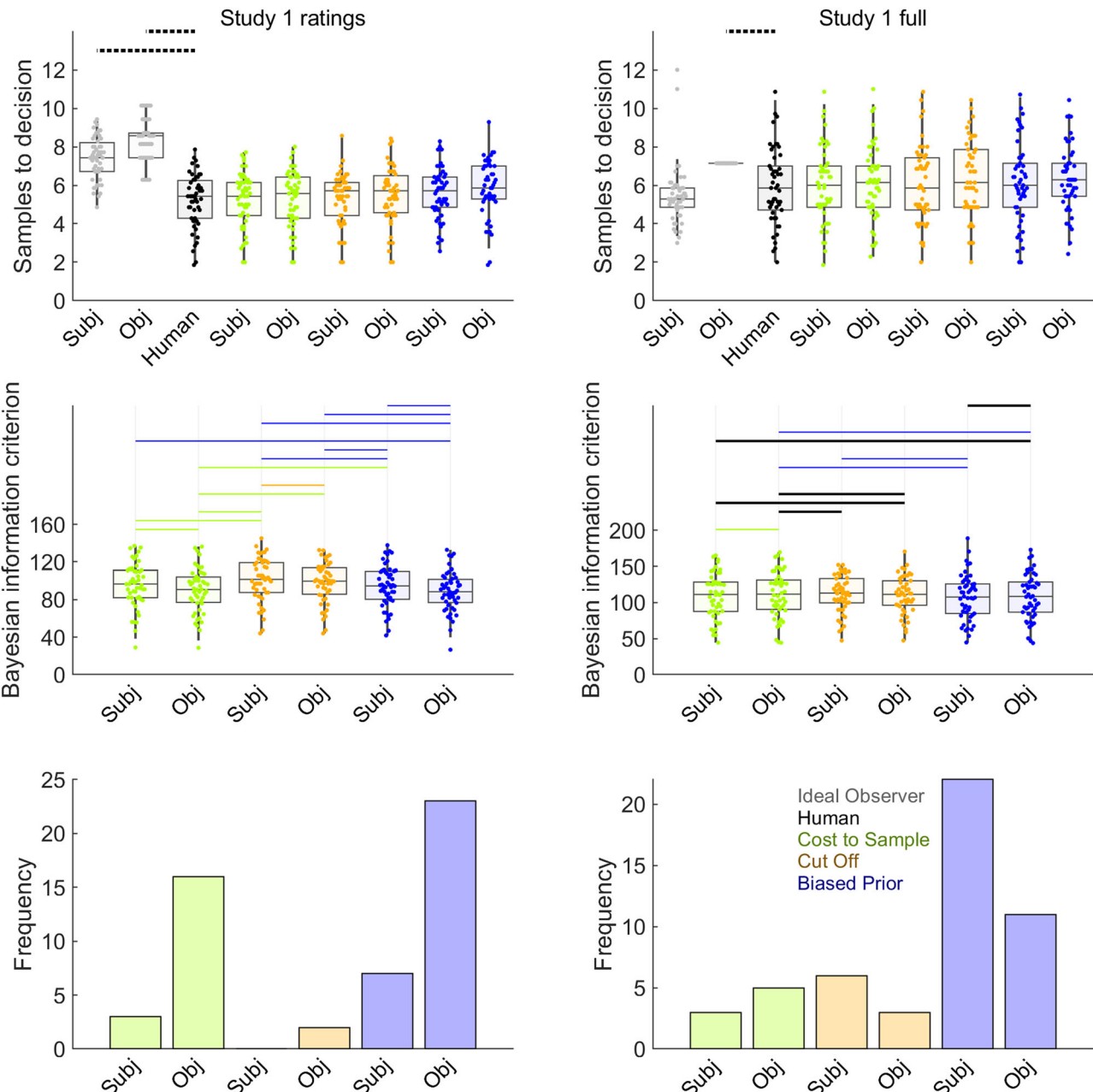

**Fig. 5 | Model comparison for Study 1 conditions with a first phase.** Results from the ratings condition ($n = 51$ participants) are shown in the left column and from the full condition ($n = 50$) on the right. Human participant sampling rates in the first row are reproduced from Fig. 2. In the first row, horizontal solid lines link samples data for humans (black points) and Ideal Observer (grey points) when $BF_{01} > 3$ (at least moderate evidence for equal means) while dotted lines indicate when $BF_{10} > 3$ (at least moderate evidence for different means). Human and Ideal Observer sampling never showed $BF_{01} > 3$ (at least moderate evidence for equal means). Statistical details for these pairwise tests can be found in Supplementary Table 1. There is clearer evidence for participant undersampling in the ratings than in the full condition. The second row shows BIC values where lower values indicate better model

fit. Horizontal lines are shown in the colour corresponding to the point spread of the better-fitting model when $BF_{10} > 3$ or in black when $BF_{01} > 3$. Statistical details for these pairwise tests can be found in Table 5. The third row shows number of participants that each model best fitted. In both conditions, some version of the Biased Prior model (blue) fits better than any other model, though the ratings condition is more ambiguous. Model point spread data colours (See also legend in lower right panel): Cost to Sample (green), Cut Off (orange), Biased Prior (blue). Boxplots reflect first, second (median) and third quartiles, while whiskers reflect 1.5 interquartile range. Point spreads reflect individual participant mean values. Abbreviations: Subj = Models that make choices about subjective values; Obj = Models that makes choices about objective values.

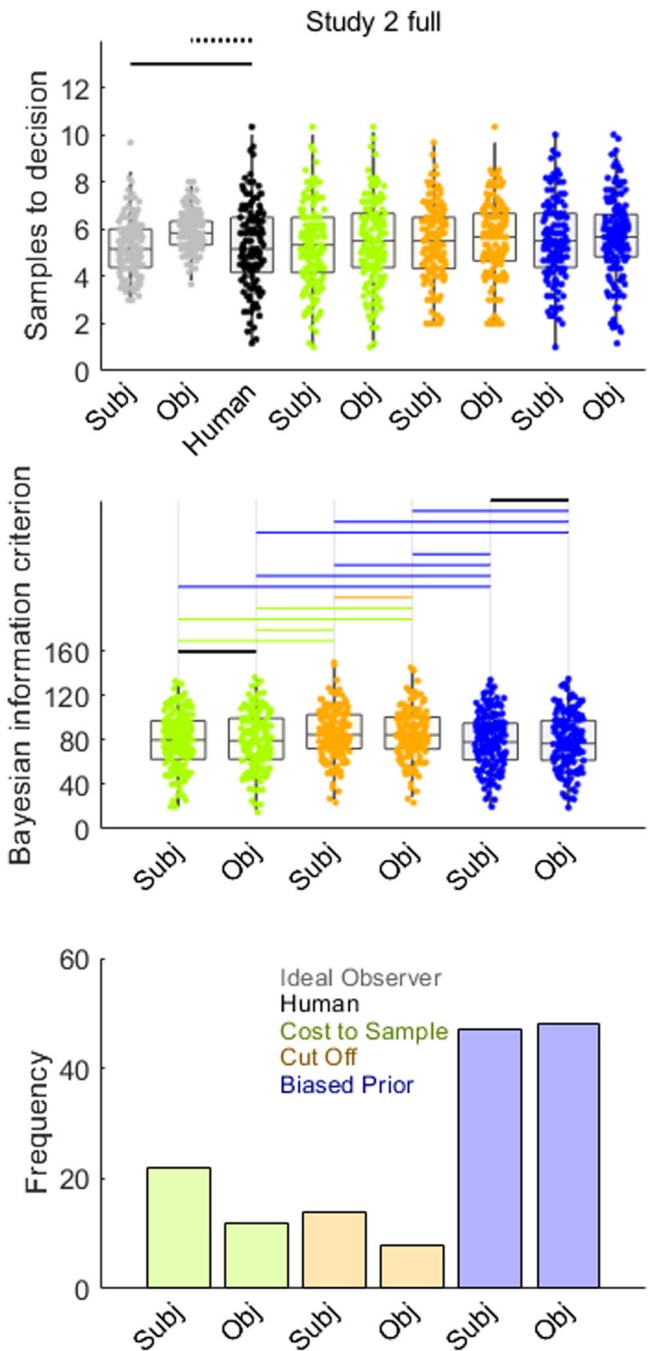

**Fig. 6 | Model comparison for Study 2.** Points in the first and second rows show sampling rates for $n = 151$ participants. Human participant sampling is reproduced from Fig. 2. In the first row, horizontal solid lines link samples data for humans (black points) and Ideal Observer (grey points) when $BF_{01} > 3$ (at least moderate evidence for equal means) while dotted lines indicate when $BF_{10} > 3$ (at least moderate evidence for different means). Statistical details for these pairwise tests can be found in Supplementary Table 1. Study 2 confirms undersampling is inconsistent at best in the full condition. The second row shows BIC values, where lower values indicate better model fit. Black horizontal lines indicate when $BF_{01} > 3$. When $BF_{10} > 3$, the horizontal line is coloured the same as the point spread of the better model. Statistical details for these pairwise tests can be found in Table 6. Biased Prior (blue) dominates other models. The third row corroborates this conclusion, as Biased Prior model also best fitted the most participants. Model point spread data colours (See also legend in lower right panel): Cost to Sample (green), Cut Off (orange), Biased Prior (blue). Boxplots reflect first, second (median) and third quartiles, while whiskers reflect 1.5 interquartile range. Point spreads reflect individual participant mean values. Abbreviations: Subj = Models that make choices about subjective values; Obj = Models that makes choices about objective values.

models (Fig. 3, middle row and Supplementary Table 2), though analysis of the number of participants best fit by each model shows dominance by the subjective values version of Biased Prior model (Fig. 3, right column, third row).

## Study 1

The discrepancy in Ideal Observer performance that we observed between Pilot baseline and Pilot full raises a distinct hypothesis, which we test in Study 1: A systematic manipulation of each of the task features added to Pilot full (Table 2) will show that at least one of them reduces Ideal Observer sampling, though none of them will modulate participant sampling. In addition to testing this hypothesis, Study 1 will give us six more datasets with which we can perform model fitting and better disambiguate whether Biased Prior best explains participant choices, including undersampling bias.

Our hypothesis was confirmed that none of the conditions affects participants' number of samples to decision (Fig. 2, with statistical detail in Supplementary Tables 5 and 6). As in our pilot studies, participants sampled slightly higher numerically in the full condition than in any other condition. However, no pair of conditions involving the full condition, nor indeed any other pair, showed a statistically substantiated mean difference either by frequentist tests (using threshold $P < 0.05$, after Bonferroni correction for the 15 condition pairs) or by Bayesian mean difference tests (using threshold $BF_{10} > 3$, at least moderate evidence in favour of mean difference). According to these Bayesian tests, nearly every pair of conditions showed statistically equivalent means (all $BF_{01} > 3$, at least moderate evidence in favour of null model, shown as magenta horizontal lines in Fig. 2), with the only exceptions being the comparisons with the full condition, which statistically showed weak or inconclusive evidence.

Even though participants' sampling rates were not affected by any task feature, the Ideal Observer appeared affected by our manipulation of payoff scheme in the full condition. The first row of Fig. 4 compares human and Ideal Observer sampling rates from the four conditions with no first phase: baseline, squares, timing and payoff. Note that all four conditions used a different payoff scheme than Study 1 full: Participants were instructed to try to choose one of the top three ranked options in each sequence. Using Bayesian pairwise tests (threshold $BF_{10} > 3$, moderate evidence for different means), we compared participants' sampling (black points in Fig. 3) in these four conditions against that of the objective values version of the Ideal Observer (grey points in Fig. 3). All four conditions showed nearly identical undersampling bias. See Supplementary Table 1 for further statistical detail for these pairwise tests.

The first row of Fig. 5 shows comparisons of participant and Ideal Observer sampling in the two conditions with an initial rating phase: ratings and full. Further statistical detail for these comparisons is found in Supplementary Table 1. Study 1 ratings (left column), a condition in which the top three ranks are rewarded, showed robust participant undersampling

Biased Prior models had statistically equal mean BIC values, but both outperformed the Cut Off heuristic in terms of mean BIC. Similarly, participant frequencies of best-fits for each model (Fig. 3, left column, third row) nearly excluded Cut Off altogether but did not well-differentiate Cost to Sample and Biased Prior models.

For Pilot full, our hypothesis was fulfilled that at least one of the several additional task features associated with oversampling[15,16] would eliminate the undersampling bias observed in Pilot baseline. This contrast between pilot studies does not seem to arise because participants sampled differently, but rather because the Ideal Observer sampled less in Pilot full, compared to Pilot baseline. In Pilot full, participants' sampling (Fig. 3, top right and Supplementary Table 1) was statistically equivalent to sampling for the objective values version of the Ideal Observer and even significantly greater than sampling for the subjective values version of the Ideal Observer. Statistical comparisons of mean BIC values did not statistically differentiate

compared to both versions of Ideal Observer. In contrast, in the full condition (right column), there was inconclusive statistical evidence for any undersampling compared to the subjective values version of the Ideal Observer. Relative to the objective value version, some statistically significant undersampling survived, though its effect size Cohen's $d = -0.61$ was cut in half compared its Study 1 ratings counterpart Cohen's $d = -1.72$. Thus, compared to all other Study 1 conditions, only the Study 1 full condition showed some degree of reduced sampling by the Ideal Observer. Note that Study 1 full is also the only condition in Study 1 where participants and the Ideal Observer were instructed to maximise the option value of their choices, instead of using a scheme that rewards only the top three ranked options.

We also evaluated computational theoretical models that could explain biases in individual participants. All the conditions produced similar results. Figure 4 and Table 4 report theoretical model comparisons for the four conditions with no first phase. All conditions replicate unambiguous evidence favouring the Biased Prior model, based on both statistical mean difference tests on BIC values (middle row, Fig. 4) and frequencies of best-fitted participants (lower row, Fig. 4). Figure 5 and Table 5 report theoretical model comparisons for the two conditions with a first phase. Study 1 ratings (Fig. 5, left) favoured the objective values version of the Bias prior model (with some contribution from Cost to Sample, objective values). Meanwhile, Study 1 full replicated Pilot full (Fig. 3) by favouring the subjective values version of the Biased Prior model (Fig. 5, right column, second and third rows), though with stronger statistical evidence on the mean BIC measure (second row) than in Pilot full. Overall, the evidence across all conditions in Study 1 collectively favours some version of the Biased Prior model as the most common explanation of participants' sampling choices.

## Study 2

The two "full" conditions we investigated so far - Pilot full and the Study 1 full - showed that an optimal stopping task in which all choices are rewarded according to their value leads to (partially) reduced Ideal Observer sampling, compared to a raft of different conditions in which only the top three ranking choices were rewarded. Participants, in contrast, maintained relatively low and invariant sampling rates across all conditions. Consequently, the two full conditions (where there was not clear undersampling) created a situation where

participants' and Ideal Observer sampling rates were close to each other, making a direct assessment of bias in this condition difficult to determine with high precision. Indeed, Pilot full and Study 1 full conflict in the biases they show relative to the objective and subjective values versions of the Ideal Observer. In Study 2 we obtained a higher quality estimate of participant sampling bias in the full condition by overcoming some limitations of our previous designs. We increased the target sample size from ~50 (in Pilot full and Study 1 full) to 151 in Study 2 full. Additionally, we generated a new set of sequence option values for every participant, whereas in Pilot full and Study 1 full, all participants engaged with sequences that were fixed in advance. These design elements also provide largest dataset for theoretical model fitting yet.

Participants appeared to sample about as many prices in Study 2 as in the previous studies reported herein (Fig. 2). Figure 6 and Supplementary Table 1 report Bayesian pairwise mean difference test results comparing participants' sampling to that of the two versions of the Ideal Observer ($BF_{01} > 3$, at least moderate evidence for null model). Here, we see that the results resemble those of Study 1 full. Participants sampled statistically equivalently to the subjective values version of the Ideal Observer (this comparison was inconclusive in the lessor-powered Study 1 full) and undersampled compared to the objective values version (replicating Study 1 full). Model fitting in Study 2 unambiguously favoured both subjective and objective values versions of the Biased Prior model on both measures of mean BIC (Fig. 6, second row, and Table 6) and frequency of best-fitted participants (Fig. 6, third row).

## Study 3

Figure 2 suggests that participants are loath to change how much they sample. They are not sensitive to the presence or absence of the various methods features listed in Table 1. And, even though rewarding only the top three options leads the Ideal Observer to increase the number of options it samples, participants do not correspondingly increase how much they sample under this incentivisation scheme. The goal of Study 3 was to ensure that our implementation of the optimal stopping task was methodologically viable and that it is in practice possible to experimentally modulate how much participants sample at least to some degree. Costa and Averbeck[8] manipulated the sequence length (i.e. how many options were available in each sequence) and

**Table 3 | Details of pairwise comparisons between participant sampling rates in Studies 1 and 3**

| Condition 1 | n | Condition 2 | n | Bayes Factor 10 (BF10) | | | t test | Cohen's d | Confidence interval | Brown-Forsythe |
|---|---|---|---|---|---|---|---|---|---|---|
| | | | | $r = 0.5$ | $r = 0.71$ | $r = 1$ | | | | |
| Study 1 | | | | | | | | | | |
| Payoff | 51 | Timing | 50 | 0.28 | 0.21 | 0.15 | $t(99) = 0.07, p = 1$ | 0.01 | −0.77 | 0.81 | $F(1,99) = 0.36, p = 1$ |
| Payoff | 51 | Squares | 50 | 0.28 | 0.21 | 0.15 | $t(99) = -0.07, p = 1$ | −0.01 | −0.8 | 0.76 | $F(1,99) = 0.60, p = 1$ |
| Payoff | 51 | Baseline | 50 | 0.29 | 0.22 | 0.16 | $t(99) = 0.33, p = 1$ | 0.07 | −0.64 | 0.79 | $F(1,99) = 4.25, p = 0.63$ |
| Payoff | 51 | Full | 50 | 0.68 | 0.55 | 0.42 | $t(99) = -1.47, p = 1$ | −0.29 | −1.49 | 0.51 | $F(1,99) = 6.16, p = 0.22$ |
| Payoff | 51 | Ratings | 51 | 0.4 | 0.31 | 0.23 | $t(100) = 0.92, p = 1$ | 0.18 | −0.56 | 1.06 | $F(1,100) = 0.02, p = 1$ |
| Timing | 50 | Squares | 50 | 0.29 | 0.21 | 0.16 | $t(98) = -0.14, p = 1$ | −0.03 | −0.81 | 0.74 | $F(1,98) = 0.02, p = 1$ |
| Timing | 50 | Baseline | 50 | 0.29 | 0.22 | 0.16 | $t(98) = 0.25, p = 1$ | 0.05 | −0.65 | 0.77 | $F(1,98) = 1.65, p = 1$ |
| Timing | 50 | Full | 50 | 0.73 | 0.59 | 0.46 | $t(98) = -1.53, p = 1$ | −0.31 | −1.51 | 0.49 | $F(1,98) = 8.18, p = 0.08$ |
| Timing | 50 | Ratings | 51 | 0.38 | 0.29 | 0.22 | $t(99) = 0.85, p = 1$ | 0.17 | −0.58 | 1.04 | $F(1,99) = 0.18, p = 1$ |
| Squares | 50 | Baseline | 50 | 0.3 | 0.23 | 0.17 | $t(98) = 0.42 p = 1$ | 0.08 | −0.59 | 0.79 | $F(1,98) = 1.33, p = 1$ |
| Squares | 50 | Full | 50 | 0.65 | 0.52 | 0.4 | $t(98) = -1.43, p = 1$ | −0.29 | −1.46 | 0.52 | $F(1,98) = 9.24, p = 0.05$ |
| Squares | 50 | Ratings | 51 | 0.43 | 0.33 | 0.25 | $t(99) = 1.01, p = 1$ | 0.2 | −0.53 | 1.06 | $F(1,99) = 0.35, p = 1$ |
| Baseline | 50 | Full | 50 | 1.09 | 0.91 | 0.71 | $t(98) = -1.82, p = 1$ | −0.36 | −1.51 | 0.37 | $F(1,98) = 17.35, p = 0.00$ |
| Baseline | 50 | Ratings | 51 | 0.34 | 0.26 | 0.19 | $t(99) = 0.70, p = 1$ | 0.14 | −0.56 | 0.9 | $F(1,99) = 2.99, p = 1$ |
| Full | 50 | Ratings | 51 | 2 | 1.73 | 1.4 | $t(99) = 2.19, p = 0.46$ | 0.44 | −0.28 | 1.75 | $F(1,99) = 6.05, p = 0.24$ |
| Study 3 | | | | | | | | | | |
| 14 options | 65 | 10 options | 75 | 1.55 | 1.29 | 1.01 | $t(138) = 2.08, p = 0.04$ | 0.35 | 0.03 | 1.33 | $F(1,138) = 3.55, p = 0.06$ |

All $p$ values and alpha values used in confidence intervals are Bonferroni corrected for 15 tested pairs. $r$ values in the Bayes Factor columns represent the scale factor of the Cauchy prior on the effect size. Compare to Fig. 2.

found participants were willing to increase the number of samples for longer sequences. Nevertheless, they also found that undersampling was more pronounced at higher sequence lengths. Participants in their study appeared reluctant to increase how much they sampled, whereas the Ideal Observer increased its sampling rate to adapt to the longer sequence lengths without constraint—a pattern that appears consistent with the reluctance with which participants increase their sampling rates in our studies reported herein. In our third study, we replicated this effect of sequence length on participants' average number of samples, using sequence lengths of 10 and 14 options. We also took the opportunity to further replicate and bolster our conclusion that a biased prior is a worthy explanation of participants' performance, using the two more datasets Study 3 provides.

Our pre-registration formally stipulated a frequentist $t$-test to test our *a priori* prediction that participants would sample more for 14 option

sequences than 10 option sequences. Indeed, Fig. 2 (rightmost boxplots) and Table 3 confirm our pre-registered prediction. The remaining analyses we now report for Study 3—comparisons of participants to Ideal Observer and analysis of theoretical model fits—go beyond our pre-registration. As in Costa and Averbeck[8], we observed undersampling at longer but not shorter sequence lengths. Indeed, the Bayesian mean difference tests (Fig. 7, upper panel, and Table 7) suggest that, at a sequence length of 10 options, participants slightly *over*sampled (rather than undersampled) compared to the subjective values version of the Ideal Observer, while the difference with objective values version remained inconclusive. In contrast, at sequence length 14, participants undersampled compared to both Ideal Observer versions. Our model-fitting (Fig. 7, middle and lower panels and Table 7) also confirmed our hypothesis that participants' sampling biases could be explained best by the Biased Prior model, though the Cost to Sample model

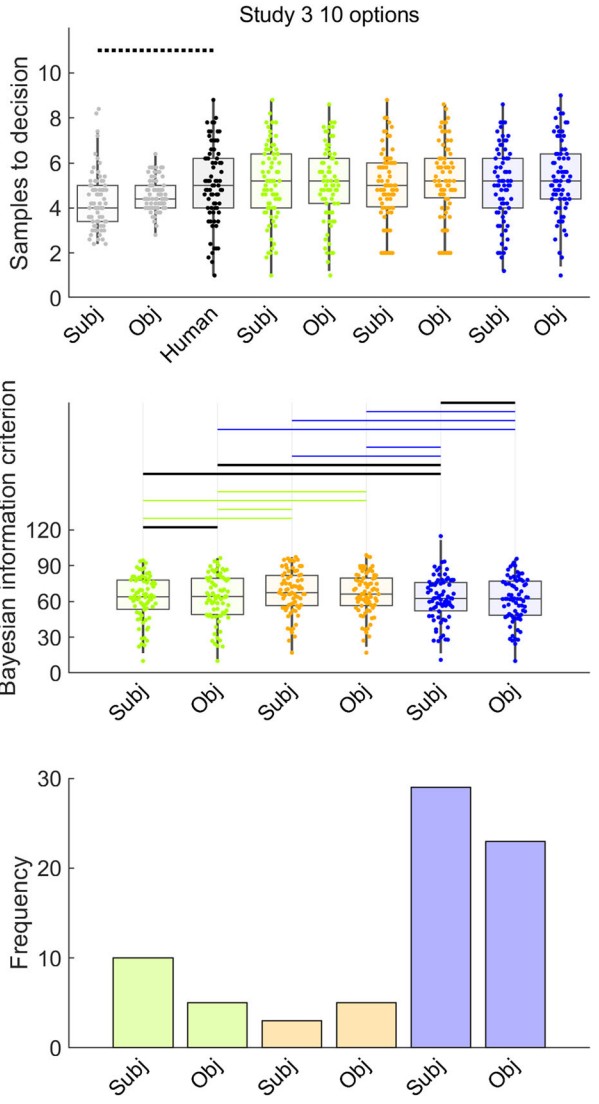

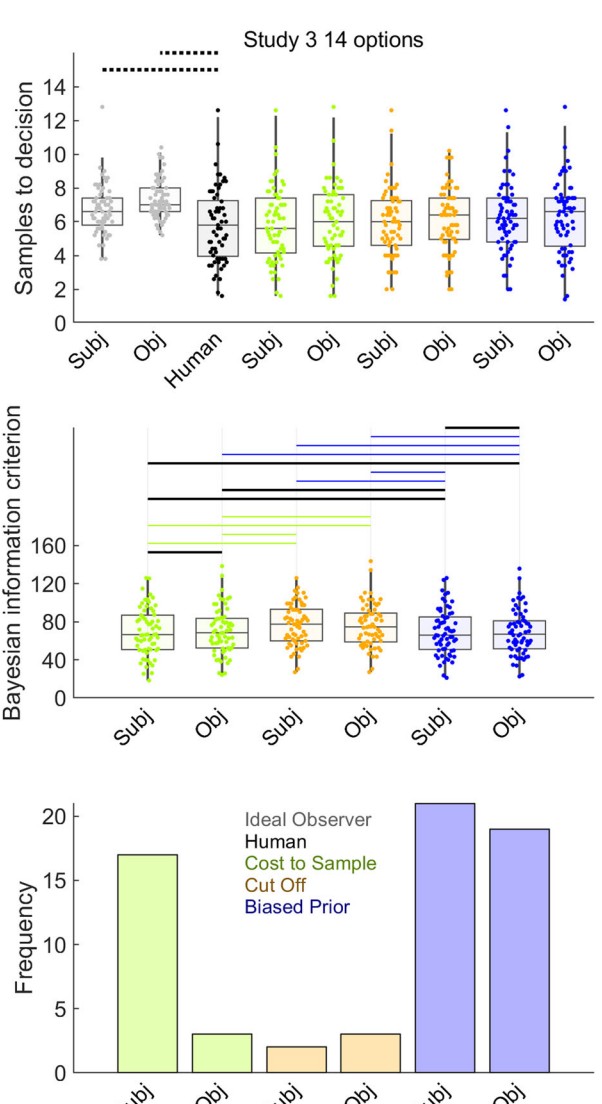

**Fig. 7 | Model comparison for Study 3.** Points in the first and second rows show sampling rates. Human participant sampling data are reproduced from Fig. 2 (See also Table 3) and show that participants sample more ($P < 0.05$) in the 14-option condition ($n = 65$ participants) than the 10-option condition ($n = 75$ participants). In the first row, horizontal solid lines link samples data for humans (black points) and Ideal Observer (grey points) when $BF_{01} > 3$ (at least moderate evidence for equal means) while dotted lines indicate when $BF_{10} > 3$ (at least moderate evidence for different means). Neither humans nor the Ideal Observer showed $BF_{01} > 3$ (at least moderate evidence for equal means). Statistical details for these pairwise tests can be found in Supplementary Table 1. Undersampling is presenting only for the longer sequence length. The second row shows BIC values, where lower values indicate

better model fit. Black horizontal lines indicate when $BF_{01} > 3$. When $BF_{10} > 3$, the horizontal line is coloured the same as the point spread of the better model. Statistical details for these pairwise tests can be found in Table 7. The third row shows numbers of participants each model best fit. The second and third rows both show favourable evidence for Biased Prior as the best model. Model point spread data colours (See also legend in lower right panel): Cost to Sample (green), Cut Off (orange), Biased Prior (blue). Boxplots reflect first, second (median) and third quartiles, while whiskers reflect 1.5 interquartile range. Point spreads reflect individual participant mean values. Abbreviations: Subj = Models that make choices about subjective values; Obj = Models that makes choices about objective values.

**Table 4 | Details of tests between pairs of mean BIC values in Study 1 models with no first phase**

| Model 1 | Model 2 | n | Bayes Factor 10 (BF10) | | | Cohen's d |
|---|---|---|---|---|---|---|
| | | | r = 0.5 | r = 0.71 | r = 1 | |
| **Study 1 baseline** | | | | | | |
| Biased Prior obj | Cut Off obj | 50 | 5.51E9 | 6.86E9 | 7.86E9 | −1.34 |
| Biased Prior obj | Cost to Sample obj | 50 | 4.5 | 3.81 | 3.03 | −0.38 |
| Cut Off obj | Cost to Sample obj | 50 | 1.18E6 | 1.35E6 | 1.40E6 | 0.98 |
| **Study 1 squares** | | | | | | |
| Biased Prior obj | Cut Off obj | 50 | 3.35E10 | 4.2210 | 4.91E10 | −1.42 |
| Biased Prior obj | Cost to Sample obj | 50 | 2085.22 | 2142.51 | 1981.89 | −0.7 |
| Cut Off obj | Cost to Sample obj | 50 | 5.22E4 | 5.72E4 | 5.62E4 | 0.84 |
| **Study 1 timing** | | | | | | |
| Biased Prior obj | Cut Off obj | 50 | 9.48E9 | 1.19E10 | 1.37E10 | −1.36 |
| Biased Prior obj | Cost to Sample obj | 50 | 10.32 | 9.05 | 7.36 | −0.43 |
| Cut Off obj | Cost to Sample obj | 50 | 1.90E4 | 2.05E4 | 1.98E4 | 0.8 |
| **Study 1 payoff** | | | | | | |
| Biased Prior obj | Cut Off obj | 51 | 2.48E14 | 3.26E14 | 4.03E14 | −1.8 |
| Biased Prior obj | Cost to Sample obj | 51 | 6.99 | 6.02 | 4.83 | −0.4 |
| Cut Off obj | Cost to Sample obj | 51 | 1.36E9 | 1.67E9 | 1.88E9 | 1.26 |

All $p$ values Bonferroni corrected for the number of tested pairs per study. $r$ values in the Bayes Factor columns represent the scale factor of the Cauchy prior on the effect size. Compare to the middle row of Fig. 4. Abbreviations: Subj = Models that make choices about subjective values; Obj = Models that make choices about objective values.

clearly made a stronger contribution in the Study 3 14 option condition than in Study 2.

## Discussion
### Undersampling bias
In our pilot studies, we first established that we could replicate an undersampling bias[4,8,12] by adapting a previous implementation of an economic full information problem[8]. In addition to this replication, we also tested novel task variables used in other studies[15,16] that hypothetically might modulate undersampling bias. We were able to modulate the size of the undersampling bias in two ways: by manipulating the payoff scheme and by manipulating sequence length .

Across four studies, we implemented so-called "full" conditions: the only condition (of many conditions that we tested) where participants maximised the option value of their choices, rather than attempted to obtain one of the top three ranked options. Every condition except these full conditions replicated robust participant undersampling, when compared to either objective or subjective values versions of the Ideal Observer. In contrast, in full conditions using 12 options or fewer, undersampling was inconsistent. This contrast between full and non-full conditions was not because participants changed their sampling rate much. It was because the Ideal Observer sampled less in the full conditions.

The Ideal Observer reduced its sampling rate in the full condition because of its payoff scheme. The full condition also implemented other task methods not present in Costa AND Averbeck[8], though we were able to experimentally eliminate these as alternative possible causes of undersampling (i.e. screen timing, grey squares, extrinsic monetary reward, the presence of a first rating phase and the use of subjective or objective values in the Ideal Observer). Thus, we must conclude that the Ideal Observer is willing to increase its sampling rate to the one appropriate for its payoff scheme (when the top three ranks are rewarded), while participants are not so willing (as they tend to always sample at nearly the same rate).

We observed a similar phenomenon in Study 3 for sequence length. Although both participants and the Ideal Observer increased their sampling rates for longer sequences (14 options compared to 10), the Ideal Observer showed a greater sampling increase for longer sequences than participants did, and thus an undersampling bias correspondingly emerged for longer

lengths but was absent for shorter ones—a finding replicated from Costa and Averbeck[8]. It appears that, while sometimes participants can increase their sampling, they generally prefer to limit how much they sample, even when it is optimal to increase sampling rate more than they do.

### Theoretical modelling
We were able to theoretically explain, in terms of a computational mechanism, participants' sampling bias. Our model fits suggest that participants' reluctance to increase sampling rates when it is optimal to do so arises because participants expect future option values to be lower on average than the ground truth. Participants of course still make some suboptimal decisions in full conditions, where undersampling bias was reduced or absent, and our data suggests that misspecified prior expectations may best account for erroneous decisions when they occur in these conditions as well. We should note, however, that the Cost to Sample model was the best-fitting model and may explain suboptimal decisions for many participants. In the Cost to Sample model, participants perceive sampling itself to be intrinsically costly or rewarding, even though there were no ground truth extrinsic costs or rewards associated with sample choices. In the case of undersampling, participants using this strategy would settle for an earlier option in part because continued sampling would be perceived as aversive. Indeed, all three of our models accurately predicted participants' mean sampling rates (Figs. 3–7). The framework we promote here, therefore—using an optimality model to explain accurate performance and then parameterising it to account for systematic bias—appears to produce models that predict participant behaviour with reasonable accuracy. It is certainly possible that different participants within the same sample might adopt any of these strategies, even if the Biased Prior strategy might be the most common.

Our study alone cannot explain whence this biased prior arises and opens new questions for future research. One possibility is that participants develop from the outside world a pre-conceived idea of the distribution of outcomes and new learning within the task (either from the ratings phase or from the sequence options themselves) fails to overwrite this preconception. Indeed, the Biased Prior model appeared to garner replicated evidence

**Table 5 | Details of tests between pairs of mean BIC values in Study 1 models with a first phase**

| Model 1 | Model 2 | n | Bayes Factor 10 (BF10) | | | Cohen's d |
|---|---|---|---|---|---|---|
| | | | r = 0.5 | r = 0.71 | r = 1 | |
| Study 1 ratings | | | | | | |
| Biased Prior obj | Biased Prior subj | 51 | 2593.14 | 2667.58 | 2469.76 | −0.7 |
| Biased Prior obj | Cut Off obj | 51 | 1.13E11 | 1.43E11 | 1.68E11 | −1.45 |
| Biased Prior obj | Cut Off subj | 51 | 7.54E11 | 9.65E11 | 1.15E12 | −1.53 |
| Biased Prior obj | Cost to Sample obj | 51 | 0.97 | 0.77 | 0.58 | −0.26 |
| Biased Prior obj | Cost to Sample subj | 51 | 5.85E5 | 6.63E5 | 6.76E5 | −0.93 |
| Biased Prior subj | Cut Off obj | 51 | 37.68 | 34.52 | 29.01 | −0.5 |
| Biased Prior subj | Cut Off subj | 51 | 4.16E7 | 4.95E7 | 5.35E7 | −1.11 |
| Biased Prior subj | Cost to Sample obj | 51 | 43.92 | 40.45 | 34.11 | 0.51 |
| Biased Prior subj | Cost to Sample subj | 51 | 2.45 | 2.02 | 1.57 | −0.34 |
| Cut Off obj | Cut Off subj | 51 | 29.43 | 26.74 | 22.33 | −0.49 |
| Cut Off obj | Cost to Sample obj | 51 | 3.11E8 | 3.78E8 | 4.18E8 | 1.19 |
| Cut Off obj | Cost to Sample subj | 51 | 2.35 | 1.94 | 1.51 | 0.33 |
| Cut Off subj | Cost to Sample obj | 51 | 3.02E9 | 3.73E9 | 4.23E9 | 1.29 |
| Cut Off subj | Cost to Sample subj | 51 | 1.19E6 | 1.36E6 | 1.40E6 | 0.96 |
| Cost to Sample obj | Cost to Sample subj | 51 | 3554.51 | 3681.92 | 3430.71 | −0.72 |
| Study 1 full | | | | | | |
| Biased Prior obj | Biased Prior subj | 50 | 0.27 | 0.2 | 0.15 | 0.11 |
| Biased Prior obj | Cut Off obj | 50 | 0.83 | 0.65 | 0.49 | −0.25 |
| Biased Prior obj | Cut Off subj | 50 | 3.07 | 2.56 | 2.01 | −0.35 |
| Biased Prior obj | Cost to Sample obj | 50 | 4.90E6 | 5.73E6 | 6.05E6 | −1.04 |
| Biased Prior obj | Cost to Sample subj | 50 | 0.32 | 0.24 | 0.17 | −0.13 |
| Biased Prior subj | Cut Off obj | 50 | 1 | 0.79 | 0.6 | −0.27 |
| Biased Prior subj | Cut Off subj | 50 | 3.57 | 3 | 2.37 | −0.36 |
| Biased Prior subj | Cost to Sample obj | 50 | 5.32 | 4.55 | 3.63 | −0.39 |
| Biased Prior subj | Cost to Sample subj | 50 | 0.61 | 0.47 | 0.35 | −0.22 |
| Cut Off obj | Cut Off subj | 50 | 1.24 | 1 | 0.76 | −0.29 |
| Cut Off obj | Cost to Sample obj | 50 | 0.31 | 0.23 | 0.17 | −0.13 |
| Cut Off obj | Cost to Sample subj | 50 | 0.34 | 0.25 | 0.19 | 0.15 |
| Cut Off subj | Cost to Sample obj | 50 | 0.21 | 0.15 | 0.11 | 0.01 |
| Cut Off subj | Cost to Sample subj | 50 | 1.13 | 0.9 | 0.69 | 0.28 |
| Cost to Sample obj | Cost to Sample subj | 50 | 5.07 | 4.32 | 3.45 | 0.39 |

All *p* values Bonferroni corrected for the number of tested pairs per study. *r* values in the Bayes Factor columns represent the scale factor of the Cauchy prior on the effect size. Compare to middle row of Fig. 5. Abbreviations: Subj = Models that make choices about subjective values; Obj = Models that make choices about objective values.

across datasets whether participants had the opportunity to learn the prior distribution from a preceding ratings task (e.g. Study 1 ratings condition, Fig. 5) or not (Fig. 4). Another possibility is that participants may learn the prior to some degree from the option values as they experience one sequence after another[25], though previous evidence speaks against such a phenomenon[7] and it is not clear why this strategy would lead to a pessimistic prior and undersampling. Baumann et al.[4] included a different approach from ours to using a learning phase prior to the optimal stopping task to ensure that participants were acquainted with the generating distribution. As in Lee and Courey[26], participants learned abstract mathematical density functions. Based on these, participants drew histograms of distributions, on which they received feedback to ensure their understanding. According to Goldstein and Rothschild[27], such a graphical elicitation technique can lead to rather accurate representations of probability distributions in participants. This approach is not likely to be especially ecologically valid, however. Another tempting explanation for participants' apparently mis-specified expectations is that participants did not treat our task as a full information problem and did not use any prior distribution. Indeed, the cut-off heuristic

derives from a "prior-free" mathematical solution to the secretary problem, which gives optimal performance assuming that participants have no knowledge of the prior distribution. Nevertheless, the Cut Off heuristic did not perform well in our model comparison, in contrast to the Cost to Sample and Biased Prior models which are based on the full information problem solution. More research into how participants learn option value distributions would be useful. We hypothesise that a biased prior might persist, regardless of how participants are exposed to the prior, though more study is needed to generalise beyond our study.

We were unable to reliably induce participants to oversample in the present work and instead we identified variables that merely modulate the size of undersampling bias. Nevertheless, others like van de Wouw et al.[16] have demonstrated and replicated oversampling bias. Their work, rather than presenting options as numeric prices as we did here, communicated option values using images, such as the attractiveness of faces, foods and holiday destinations. Our manipulations of task features in Study 1 have already tested and rejected other task differences used in their paradigm that might give rise to oversampling (e.g. grey squares, timing, etc), leaving the

**Table 6 | Details of tests between pairs of mean BIC values in Study 2**

| Model 1 | Model 2 | n | Bayes Factor 10 (BF10) | | | Cohen's d |
|---|---|---|---|---|---|---|
| | | | r = 0.5 | r = 0.71 | r = 1 | |
| Biased Prior obj | Biased Prior subj | 151 | 0.15 | 0.11 | 0.08 | −0.05 |
| Biased Prior obj | Cut Off obj | 151 | 9.74E16 | 1.09E17 | 1.10E17 | −0.87 |
| Biased Prior obj | Cut Off subj | 151 | 2.58E19 | 2.97E19 | 3.07E19 | −0.94 |
| Biased Prior obj | Cost to Sample obj | 151 | 96269.85 | 86925.33 | 71670.29 | −0.45 |
| Biased Prior obj | Cost to Sample subj | 151 | 2.35 | 1.79 | 1.32 | −0.2 |
| Biased Prior subj | Cut Off obj | 151 | 9.57E14 | 1.05E15 | 1.03E15 | −0.8 |
| Biased Prior subj | Cut Off subj | 151 | 2.18E19 | 2.51E19 | 2.59E19 | −0.94 |
| Biased Prior subj | Cost to Sample obj | 151 | 4.16 | 3.21 | 2.38 | −0.22 |
| Biased Prior subj | Cost to Sample subj | 151 | 7.93 | 6.19 | 4.63 | −0.24 |
| Cut Off obj | Cut Off subj | 151 | 4.75 | 3.67 | 2.73 | −0.23 |
| Cut Off obj | Cost to Sample obj | 151 | 5.08E10 | 5.22E10 | 4.79E10 | 0.66 |
| Cut Off obj | Cost to Sample subj | 151 | 1.19E11 | 1.23E11 | 1.14E11 | 0.68 |
| Cut Off subj | Cost to Sample obj | 151 | 6.78E13 | 7.32E13 | 7.05E13 | 0.77 |
| Cut Off subj | Cost to Sample subj | 151 | 7.29E14 | 7.98E14 | 7.79E14 | 0.8 |
| Cost to Sample obj | Cost to Sample subj | 151 | 0.21 | 0.15 | 0.11 | 0.08 |

All p values Bonferroni corrected for the number of tested pairs per study. r values in the Bayes Factor columns represent the scale factor of the Cauchy prior on the effect size. Compare to middle row of Fig. 6. Abbreviations: Subj = Models that make choices about subjective values; Obj = Models that make choices about objective values.

pictorial stimulus domains as the most likely instigator of oversampling. It is yet unknown, but possible, that a biased (i.e. overly optimistic) prior might account for oversampling in image-based contexts as well as undersampling in number-based, economic domains.

We have built and comprehensively compared multiple theoretical models, where we aimed to specify the computations humans use to solve full information problems. Costa and Averbeck[8] have previously reported the Cost to Sample model that we consider here and fitted that model to participants' sampling choices in an economic full information task. However, they did not perform a model comparison with alternative models. Moreover, our current study provides comprehensive parameter and model recovery analyses for this Cost to Sample model and four other theoretical models. Our work elaborates on the approach taken by Baumann et al.[8], who compared the objective values version of the Cut Off heuristic we consider here with "threshold models". These threshold models assert, as our models do, that participants compare the value of each option against a threshold at each sequence position[7]. Their approach estimates decision thresholds directly from participants' choices, even if the thresholds may be subject to constraints—with a linearity constraint proving best fitting. As such, threshold models provide an excellent descriptive tool for recovering thresholds from data, and hypotheses may then be formed about what computational or psychological mechanisms might give rise to those thresholds. However, this threshold model fitting approach does not formally articulate nor test any hypothetical causes of threshold changes. That is, neither heuristic[28] nor more complex computations are proposed or shown to cause the thresholds. A similar descriptive approach fits probability distributions to choice patterns over sequence positions, where informal mechanisms might be hypothesised to cause these choice distributions[29]. Much has been learned from these approaches, though we took a different approach to our model space here. We strove to develop models that are "computational" in the sense that they specify the computations that participants deploy to solve the task, including the mechanisms by which thresholds come to be computed in a biased way. Our modelling approach is distinct from a "bias from optimal" approach[4,30], where bias parameters merely reflect differences from optimal thresholds[28]. Bias parameters like Cost to Sample or Biased Prior instead have straightforward psychological interpretations, and the models explicitly articulate formally how these bias parameters skew threshold computation.

**Limitations**

We have introduced a framework whereby optimality solutions including that of the Secretary Problem and that of the full information problem have been leveraged to explain accurate performance on optimal stopping tasks. And our framework has taken the approach of parameterising these models to explain systematic sources of suboptimal performance. Given that we have proposed this framework and demonstrated its utility, we expect that future research can refine the models we have proposed to build improved models that may better fit participant data. For example, more complex models that combine multiple bias-related free parameters (e.g. the cost to sample parameter and a constant added to the prior mean as fitted parameters at the same time) might be considered. Also, more sophisticated versions of our models might be formulated, such as cost to samples that change across sequence position.

When considering models that might be built and tested in the future, it is worth considering that the Bayesian solution to the full information problem (which we used as a base for some of our models) is relatively computationally complex, especially its backwards induction algorithm. Future research might explore models that use a more limited-capacity backwards induction, which can only partially explore possible future states, or create some simpler heuristic to approximate the choice threshold/value of sampling again. We note that already our evidence here undermines the case for a previously proposed heuristic, the Cut Off heuristic. In any case, we do not know the capacity of the neural architectures involved in solving these problems, rendering it difficult to reject models a priori on the basis of limited capacity. Indeed, it is plausible that neurons may be implementing similar computations as the kinds of models we investigated here. It has already been shown that brain responses correlate trial-by-trial with fluctuations in quantities derived from backwards induction-based optimal stopping models that have been fitted to human participant choice data[8,31].

We have already mentioned several ways that the modelling framework we propose here and the results we have obtained raise new research questions and open new research lines. One last issue that we also feel deserves further study relates to the extent to which systematic bias translates into real losses for people confronted with real optimal stopping problems. We also show herein that undersampling bias can vary widely depending on the parameters of the decision problem: factors such as sequence length and incentivisation can affect the size of bias. Just how large

**Table 7 | Details of tests between pairs of mean BIC values in Study 3**

| Model 1 | Model 2 | n | Bayes Factor 10 (BF10) | | | Cohen's d |
|---|---|---|---|---|---|---|
| | | | r = 0.5 | r = 0.71 | r = 1 | |
| Study 3 10 options | | | | | | |
| Biased Prior obj | Biased Prior subj | 75 | 0.31 | 0.23 | 0.17 | −0.13 |
| +Biased Prior obj | Cut Off obj | 75 | 6.05E6 | 6.55E6 | 6.35E6 | −0.79 |
| Biased Prior obj | Cut Off subj | 75 | 8.4E9 | 9.8E9 | 1E10 | −1 |
| Biased Prior obj | Cost to Sample obj | 75 | 1.89 | 1.49 | 1.13 | −0.27 |
| Biased Prior obj | Cost to Sample subj | 75 | 0.26 | 0.19 | 0.14 | −0.1 |
| Biased Prior subj | Cut Off obj | 75 | 695.77 | 649.11 | 551.07 | −0.52 |
| Biased Prior subj | Cut Off subj | 75 | 5.9E7 | 6.6E7 | 6.6E7 | −0.86 |
| Biased Prior subj | Cost to Sample obj | 75 | 0.19 | 0.14 | 0.1 | 0.04 |
| Biased Prior subj | Cost to Sample subj | 75 | 0.21 | 0.15 | 0.11 | 0.07 |
| Cut Off obj | Cut Off subj | 75 | 3.3 | 2.66 | 2.04 | −0.3 |
| Cut Off obj | Cost to Sample obj | 75 | 8.44E4 | 8.61E4 | 7.89E4 | 0.67 |
| Cut Off obj | Cost to Sample subj | 75 | 2.7E4 | 2.72E4 | 2.45E4 | 0.63 |
| Cut Off subj | Cost to Sample obj | 75 | 1.2E9 | 1.4E9 | 1.4E9 | 0.94 |
| Cut Off subj | Cost to Sample subj | 75 | 4.5E11 | 5.4E11 | 5.9E11 | 1.11 |
| Cost to Sample obj | Cost to Sample subj | 75 | 0.18 | 0.13 | 0.09 | 0.01 |
| Study 3 14 options | | | | | | |
| Biased Prior obj | Biased Prior subj | 65 | 2.3 | 1.85 | 1.42 | −0.3 |
| Biased Prior obj | Cut Off obj | 65 | 3.8E10 | 4.5E10 | 5E10 | −1.15 |
| Biased Prior obj | Cut Off subj | 65 | 2.1E10 | 2.5E10 | 2.7E10 | −1.13 |
| Biased Prior obj | Cost to Sample obj | 65 | 17.62 | 15.24 | 12.24 | −0.4 |
| Biased Prior obj | Cost to Sample subj | 65 | 0.83 | 0.65 | 0.48 | −0.23 |
| Biased Prior subj | Cut Off obj | 65 | 6436.71 | 6449.61 | 5817.81 | −0.64 |
| Biased Prior subj | Cut Off subj | 65 | 4.64E6 | 5.15E6 | 5.12E6 | −0.86 |
| Biased Prior subj | Cost to Sample obj | 65 | 0.22 | 0.16 | 0.12 | 0.08 |
| Biased Prior subj | Cost to Sample subj | 65 | 0.29 | 0.22 | 0.16 | 0.12 |
| Cut Off obj | Cut Off subj | 65 | 0.62 | 0.47 | 0.35 | −0.2 |
| Cut Off obj | Cost to Sample obj | 65 | 2.4E7 | 2.7E7 | 2.7E7 | 0.92 |
| Cut Off obj | Cost to Sample subj | 65 | 8.52E4 | 8.94E4 | 8.41E4 | 0.73 |
| Cut Off subj | Cost to Sample obj | 65 | 9.1E7 | 1E8 | 1.1E8 | 0.96 |
| Cut Off subj | Cost to Sample subj | 65 | 1.6E8 | 1.8E8 | 1.9E8 | 0.98 |
| Cost to Sample obj | Cost to Sample subj | 65 | 0.19 | 0.14 | 0.1 | 0 |

All p values Bonferroni corrected for the number of tested pairs per study. r values in the Bayes Factor columns represent the scale factor of the Cauchy prior on the effect size. obj = Compare to the middle row of Fig. 7. Abbreviations: Subj = Models that make choices about subjective values; Obj = Models that make choices about objective values.

biases can eventually become and the extent to which significant losses might accrue for agents due to ever larger biases cannot be answered directly by our data and would benefit from more targeted investigations. Ideally, future studies could better approximate real-world conditions of decision making, or even collect field data "from the wild", to assess what kinds of losses might (or might not) occur under more ecologically valid conditions, as opposed to the more tightly controlled studies we report here.

## Conclusion

In summary, we show that the sampling rate of the Ideal Observer (which reflects optimal performance) is relatively more sensitive than those of participants to (at least) manipulations of payoff schemes and sequence lengths, such that these two factors can modulate the degree of under-sampling bias. We explain participants' sampling behaviour using a theoretical model by which participants implement optimal Bayesian computations to solve the task accurately, but a systematic undersampling bias develops when participants mis-predict the quality of upcoming

sampling, based on biased beliefs about the probability distribution of outcomes.

## Data availability
Data are available at https://github.com/nicholasfurl/Model_fitting_hybrid_study and https://doi.org/10.5281/zenodo.14282312.

## Code availability
Code is available at https://github.com/nicholasfurl/Model_fitting_hybrid_study and https://doi.org/10.5281/zenodo.14282312.

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

## Acknowledgements

Ryan McKay acknowledges funding support from the NOMIS Foundation ("Collective Delusions: Social Identity and Scientific Misbeliefs"). The funders had no role in the conceptualization, design, data collection, analysis, decision to publish, or preparation of the manuscript. Thanks to Bruno Averbeck and Matteo Lisi for their guidance with modelling and Bruno Averbeck also for his assistance with MATLAB code for model implementation.

## Author contributions

Didrika van de Wouw contributed to study conception, design, data analysis and manuscript writing. Ryan MacKay contributed to study conception, design and manuscript writing. Nicholas Furl contributed to study conception, design, data analysis and manuscript writing.

## Competing interests

The authors declare no competing interests.
