## [Transparent Peer Review file · Communications Psychology]

Biased expectations about future choice options predict sequential economic decisions

Corresponding Author: Dr Nicholas Furl

Version 0:

Decision Letter:

Dear Dr van de Wouw,

Thank you for your patience during the peer-review process. Your manuscript titled "How we model prior belief is crucial for predicting decision biases in realistic contexts" has now been seen by 2 reviewers, whose comments are appended below. You will see that they find your work of some potential interest. However, they have raised quite substantial concerns that must be addressed. In light of these comments, we cannot accept the manuscript for publication, but would be interested in considering a revised version that fully addresses these serious concerns.

We hope you will find the Reviewers' comments useful as you decide how to proceed. Should additional work allow you to address these criticisms, we would be happy to look at a substantially revised manuscript. If you choose to take up this option, please highlight all changes in the manuscript text file, and provide a detailed point-by-point reply to the reviewers.

Editorially, we consider three aspects key: the evidence put forward in support of your interpretation must be strengthened through additional empirical data and further analyses, the key advance must be demonstrated more convincingly and explained more clearly and the use of appropriate statistics and improved statistics reporting is required. Please note that the editorial requests incorporate advice we received from Reviewer #2 in an additional email exchange in which we enquired about ways to address their key criticism as listed below.

First, as Reviewer #2 highlights, it is presently not evident whether the absence of measurable change in behaviour is a result of manipulations that are genuinely without an effect, or a feature of the paradigm. You will need to conduct additional work that demonstrates that the task allows manipulation of human behaviour. This work should be preregistered and powered a priori to detect subtle effects. Please also note Reviewer #1's critique regarding the effect of payoff schemes as you address this point.

Second, as likewise mentioned by Reviewer #2, a key issue is clarifying and strengthening the insights that arise from the work. The effects demonstrated here arise from a comparison between computational models of human behaviour, with little insight into why human behaviour differs from the optimal solution; at a minimum, the goal of revision should be to convincingly demonstrate that commonly used implementations of the model produced results that are artificially interpreted as over/undersampling.

Finally, Reviewer #1 provides a number of constructive suggestions for how additional analyses would strengthen the evidence and generate a more complete understanding of human behaviour in the task. We ask you to address these suggestions, and at the same time, provide Bayesian statistics or equivalence tests for all null-results, which can otherwise not be interpreted. You will find more information about our guidelines for statistics in the PS.

If the revision process takes significantly longer than six months, we will be happy to reconsider your paper at a later date, provided it still presents a significant contribution to the literature at that stage.

We understand that due to the current global situation, the time required for revision may be longer than usual. We would

appreciate it if you could keep us informed about an estimated timescale for resubmission, to facilitate our planning. Of course, if you are unable to estimate, we are happy to accommodate necessary extensions nevertheless.

Please use the following link to submit your revised manuscript, point-by-point response to the Reviewers' comments with a list of your changes to the manuscript text (which should be in a separate document to any cover letter) and any completed checklist:

Link Redacted

Please do not hesitate to contact me if you have any questions or would like to discuss the required revisions further. Thank you for the opportunity to review your work.

Best wishes,

Marika

Marika Schiffer, PhD
Chief Editor
Communications Psychology

EDITORIAL POLICIES AND FORMATTING

Editorial Policy: [Policy requirements](https://www.nature.com/documents/nr-editorial-policy-checklist.pdf) (Download the link to your computer as a PDF.)

Furthermore, please align your manuscript with our format requirements, which are summarized on the following checklist: [Communications Psychology formatting checklist](https://www.nature.com/documents/commsj-psychol-style-formatting-checklist-article.pdf)

and also in our style and formatting guide [Communications Psychology formatting guide](https://www.nature.com/documents/commsj-psychol-style-formatting-guide-accept.pdf) .

* **CODE AVAILABILITY:** All Communications Psychology manuscripts must include a section titled "Code Availability" at the end of the methods section. In the event of publication, we require that the custom analysis code supporting your conclusions is made available in a publicly accessible repository; please choose a repository that provides a DOI for the code; the link to the repository and the DOI must be included in the Code Availability statement. Publication as Supplementary Information will not suffice.

* **DATA AVAILABILITY:**

All Communications Psychology research manuscripts must include a section titled "Data Availability" at the end of the Methods section or main text. More information on this policy, is available at <http://www.nature.com/authors/policies/data/data-availability-statements-data-citations.pdf>.

At a minimum the Data availability statement must explain how the data can be obtained and whether there are any restrictions on data sharing. Communications Psychology strongly endorses open sharing of data. If you do make your data openly available, please include in the statement:

- Unique identifiers (such as DOIs and hyperlinks for datasets in public repositories)
- Accession codes where appropriate

- If applicable, a statement regarding data available with restrictions
- If a dataset has a Digital Object Identifier (DOI) as its unique identifier, we strongly encourage including this in the Reference list and citing the dataset in the Data Availability Statement.

We recommend submitting the data to discipline-specific, community-recognized repositories, where possible and a list of recommended repositories is provided at <http://www.nature.com/sdata/policies/repositories>.

If a community resource is unavailable, data can be submitted to generalist repositories such as [figshare](https://figshare.com/) or [Dryad Digital Repository](http://datadryad.org/). Please provide a unique identifier for the data (for example a DOI or a permanent URL) in the data availability statement, if possible. If the repository does not provide identifiers, we encourage authors to supply the search terms that will return the data. For data that have been obtained from publicly available sources, please provide a URL and the specific data product name in the data availability statement. Data with a DOI should be further cited in the methods reference section.

REVIEWERS' EXPERTISE:

Reviewer #1, decision making, computational modelling
Reviewer #2, decision making, computational modelling

REVIEWERS' COMMENTS:

Reviewer #1 (Remarks to the Author):

In this manuscript the authors examined sampling behavior in full information stopping problems. First, in a pilot study, they showed that the participants' sampling rate is lower (numerically) than that of a Bayesian ideal observer model that relied on the objective price values (undersampling bias), but higher than that of an ideal observer model that relied on the subjective values (oversampling bias). Then, in their main study, they showed that the sampling rate of the participants did not significantly change across different task features. However, the classification of the participants' performance (undersampling or oversampling) changed as a result of the model it was compared to: a comparison to model 1 (objective values) resulted in undersampling bias (or no bias at all), while a comparison to model 2 (subjective values) flipped the results. The authors suggest that this pattern of results stems from differences in the prior distributions of the objective (model 1) and the subjective values (model 2).

The results are novel and interesting and the manuscript is well-written and easy to follow. I have a few suggestions that I hope would help to improve it.

1. Payoff schemes – I found the different payoff schemes (Reward 1 & Reward) a bit problematic. The authors mention that “payoff schemes at their most potent cannot switch between under- versus over-sampling biases” (Supplementary Text C). However, the different payoff schemes can almost completely eliminate the bias. That is, to change the classification of behavior from under/oversampling bias to no bias at all.

2. Statistical backup – In several places the authors interpret the results based on visual inspection without backing up their claims with statistical analyses. For example, in Figure 4A the subjective values are presented as a function of the objective prices, and the relationship between them is described as ‘sigmoidal’. However, this relationship should be examined more quantitatively. For instance, the authors could fit a mixed model sigmoidal function to the data and compare its fit to other models (e.g., linear model, decaying exponential function, etc.). In addition, the authors mention that the prior distributions in Figure 4B are different, but do not show a significant difference between them (e.g., using a Kolmogorov–Smirnov test). Finally, in Supplementary Text C, the authors compare different conditions without performing any statistical analyses (for example, is the mean sampling rate in Model 1 – reward 1 significantly higher than the mean sampling rate in Model 1 – reward 2? etc.).

3. Additional Analyses/discussions – In my opinion, several statistical analyses/discussions could be added to better understand the data and strengthen the manuscript:

- Learning effects – Did the participants show any learning effects during the studies? For example, did they move from undersampling to oversampling across trials (or vice versa)?
- Bayesian models – Models 1 & 2 provide benchmarks to which participants' sampling rate is compared. Could the authors add a discussion (or analysis) about whether these models can also account for the cognitive mechanisms underlying the behavior of the participants.
- Sampling efficiency – It would be interesting to compare the mean payoff obtained by the participants to that obtained by the optimal models.
- Rating consistency – Each price was presented twice at the rating phase. What was the correlation between the two

ratings?

· Subjective vs. Objective values – The authors speculate that participants sampling rate would be affected by the distribution of subjective perception of prices (e.g., £550 is roughly equal to £400, p. 10/ first paragraph). This can be empirically examined by comparing participants the sampling rate of participants with a relatively linear relation between the subjective values and objective prices, to participants with a more 'curvy' relation.

· Effect Size – The authors reported only significance levels, but not effect sizes.

4. Code availability – As this manuscript compares human sampling behavior to that of ideal observer models, it would be great if the authors could make the code used to implement the ideal observers publically available.

Reviewer #2 (Remarks to the Author):

Summary

The paper investigates full information optimal stopping problems, specifically when people oversample/undersample in this scenario. The author first hypothesized that the number-based tasks led to undersampling and picture-based tasks led to oversampling (as reported by previous studies). But oversampling was observed in the number-based task in the Pilot study, which indicates that pictorial stimuli may not be the only reason causing oversampling. The goal of the Main study is to isolate which task feature leads to oversampling in number-based tasks. They found the human sampling rate is unchanged in all six conditions (i.e., Baseline, Full, Squares, Payoff, Timing, and Ratings). The conclusion about over versus undersampling is determined by the implementation of the model. The paper implemented two models with different prior generating distributions for the Bayesian optimality model. Model 1 uses objective prices as the prior generating distribution, and Model 2 uses subjective evaluations of prices.

Review

The potential contributions of the paper are to show that some task features are insignificant in affecting human sampling biases and that the conclusions about under/oversampling are completely model-based. However, the paper falls short in several places. It is not clear that the study itself was sensitive enough to detect any changes in behavior so it gives low confidence in the conclusion that human sampling is unaffected by task features. The paper does not give any insight as to why there is a difference between the Bayesian optimal models and human behavior; the only conclusion here is that participants under sample the Bayesian optimal model, but even that conclusion, as the paper establishes, is completely model-based. Finally, overall the paper was very difficult to parse. Thus, I cannot recommend the paper for publication.

Comments

1. ****Sensitivity to task features.**** The author state, "Our Pilot Study shows that oversampling is not limited to tasks that present option values as pictures but can also occur for some tasks using numeric stimuli to communicate option value. Our Main Study then attempted to systematically isolate which task feature leads to oversampling on number-based tasks." The idea here was that there were several features in the experimental protocol that apparently differed between the current numeric version and the past numeric versions that established undersampling. The Main study then changed these features to see if that could explain the difference between the current result and the past result. But, one reasonable issue here is a sensitivity issue. Could it be that in this particular study, participants are just relatively insensitive to these task features? That is, they should show that the study could actually prompt changes in behavior and that the study can detect it. That would give confidence in the conclusion that participants are insensitive to task changes.

2. ****Model 1 and Model 2**** A real struggle in this paper is the use of Model 1 and Model 2 and the definition of undersampling and oversampling. Over many, many reads. Here is what I understand. Model 1 is the Bayesian Optimal model with objective values. Model 2 is the Bayesian Optimal Model with subjective values. Many papers showed in these optimal search tasks that people under sample with references to the Bayesian Optimal Model with objective values (Model 1). A couple of papers (Furl et al., 2019; van de Wouw et al., 2022) came out using images instead of numeric values, which meant researchers needed to collect subjective values of the images to run the Bayesian Optimal Model. But, comparing behavior to this model led to the conclusion that people oversample. This paper set up a situation where both Model 1 and Model 2 could be used, establishing that the conclusion of undersampling (with reference to Model 1) and oversampling (with reference to Model 2) is completely model-based. This is useful information, but honestly, it takes a lot of effort for the reader to figure this conclusion out. The writing obscures this because it flips between Model 1 and Model 2. But, there could be a lot more work done to make that clear. For instance, in the Pilot study, it is really confusing to read that "Our Pilot Study shows that oversampling is not limited to tasks that present option values as pictures but can also occur for some tasks using numeric stimuli to communicate option value. Our Main Study then attempted to systematically isolate which task feature leads to oversampling on number-based tasks." But, as a reader, you are looking at Figure 1 and Model 1 and saying there

is no oversampling.

A deeper issue here is that, in the end, the main conclusion of the paper is that the conclusion of over vs undersampling is completely model-based. But, this is not that interesting because it is true by definition. There is really little to no insight as to why human behavior differs from the optimal math solution. The paper's opening suggests it is going to investigate this, but the studies do not do this. The main study seems to have been designed to do this, but really all we learn is that changes in the experimental task features did not have any impact. And as mentioned above, there is a real issue here that it could be that the experiment itself just can't show or detect any changes in human behavior.

The paper could also generate models that closely replicate human stopping behavior and find the best model by comparing the models' results with human behavior. The authors seemed to try to do this with, for instance, the subjective value version of the Bayesian model. But, then, the experiments give us no information about why there are differences between human behavior and the models. Moreover, it isn't really clear if the Bayesian optimal models would have predicted any changes based on the manipulations in the main study.

Minor Comments

1. The difference between 1 and 10 is not the same as between 1000 and 1010.
2. "Although full information problems lack the many restrictive assumptions of the secretary problem, they cannot employ such a simple rule to derive optimal performance for comparison with human performance. "
3. The paper states that secretary problem is a simpler problem of full information problems, and the simple mathematical rule of the secretary problem cannot be used to derive optimal performance for comparison with human performance. But the secretary problem is known as the "no-information game", the assumptions of the no information game are very different from those of the full information game. Thus, the optimal solution of no information game is of course not applicable to the full information game.

Version 1:

Decision Letter:

Dear Nick,

Thank you for your patience during the peer-review process. Your manuscript titled "Biased expectations about future choice options predict sequential economic decisions" has now been seen by the same 2 reviewers as before, and I include their comments at the end of this message. They find your work of interest but raised some important points. We remain interested in the possibility of publishing your study in Communications Psychology, but would like to consider your responses to these concerns and assess a revised manuscript before we make a final decision on publication.

We therefore invite you to revise and resubmit your manuscript, along with a point-by-point response to the reviewers. Please highlight all changes in the manuscript text file.

Editorially, we consider the following points key: both reviewers request a qualitative evaluation of the model fit that demonstrates how well the model emulates human data. Second, the reviewers raise questions about the (cognitive) feasibility of the modelled processes. Finally, they raise the question of the consequences of (non)optimal behaviour on the task.

The referees also highlight the potential for more model comparisons (additional models). While this may be interesting for future work, we ask you to not place an emphasis on this issue, and rather focus on the requests outlined above that will aid to demonstrate the characteristics and plausibility of the models currently applied.

I am attaching an Editorial Requests Table that details critical reporting requirements for the revised manuscript. Please attend to each item and ensure your manuscript is fully compliant. We are requesting that your manuscript aligns with these requirements as this facilitates the evaluation of your manuscript, reducing delays in re-review and potential future acceptance. If your revised manuscript is not aligned with these requests on major issues, such as those concerning statistics, it may be returned to you for further revisions without re-review. Additional information can be found in our style and formatting guide <https://www.nature.com/documents/commspsychol-style-formatting-guide-accept.pdf> >Communications Psychology formatting guide.

Please use the following link to submit your

- revised manuscript,
- point-by-point response to the referees' comments,
- cover letter (as a separate document),
- the Editorial Policy Checklist (see below),
- the Reporting Summary (see below), and
- the completed Editorial Request Table (attached):

Link Redacted

Best regards,

Marike

Marike Schiffer, PhD
Chief Editor
Communications Psychology

REVIEWER REPORTS:

Reviewer #1 (Remarks to the Author):

I thank the authors for addressing my previous comments and for the considerable effort invested in revising the manuscript.

In this revised version of the manuscript, the authors have added several theoretical models aimed at explaining the behavior of the participants. Generally, the addition of these models improved the manuscript. However, since the article has undergone considerable changes, I have a few additional comments regarding the new version of the manuscript, both in relation to its clarity and in relation to the newly added models.

- Clarity and presentation - The introduction and paradigm description remain relatively clear. However, the model description and results sections are relatively complex and less intuitive. I suggest several improvements:

- o Ideal observer model - The basic model is explained in a relatively abstract way. A simple numerical example or visualization (e.g., one that illustrates the backwards induction) could help readers who are not familiar with this model to understand it better.

- o Abbreviations - Throughout the article, the authors used abbreviations such as CO, CS, etc. Using the full names of the models could enhance readability and flow of the text.

- o A table summarizing the key features of each model could help.

- o Figure 1 - The font size is relatively small, making it difficult to see what is written.

- o Figures 2-7 - i. Adding a legend explaining the color coding would assist in understanding the Figure more easily. ii. Some of the BIC values look almost identical, and it is very difficult to distinguish between them. Adding numerical BIC values would facilitate easier comparisons between the models. iii. Figure 3 - Column headings would make the figure clearer.

- Qualitative versus quantitative comparison - While the BP model's superior BIC values indicate a better fit, it would be helpful to understand the specific qualitative aspects of the data that contribute to this superiority. For example, how does the BP model's prediction of participant behavior differ from the CS model (i.e., both models predict under-sampling, so where are their predictions different)? Providing a more detailed comparison of the predictions and behavior patterns could illuminate why the BP model outperforms others.

- Additional model considerations - The models implemented provide valuable insights, but exploring additional models could yield even richer findings.

- o For example, did the authors examine a combined BP and CS model? Since the mechanisms underlying these two models are different, a combined model might be able to achieve better results.

- o Another possible model is one similar to BP/CS but assumes time-varying parameters (for example, that the sampling cost increases over time). This idea is similar to the urgency signal/collapsing boundaries in sequential sampling models.

- o The CS and BP models are essentially versions of the IO model. There may be models from different families that might explain the participants' undersampling. For example, a model that assumes that the participants scanned the space of possibilities partially (e.g., only for 3-4 levels), and therefore undersampled.

- o Exploring some of these models (or at least discussing them as potential avenues in the discussion section) could strengthen the manuscript.

- Model feasibility - In the context of the previous comment, the BP model outperformed the other models in terms of BIC results. However, the feasibility of the calculation required by it raises questions about its psychological plausibility. How realistic is it for participants to perform such complex calculations? Could simpler heuristics underlie what appears to be complex computation?

- Optimality - Given that human performance closely matched the optimal model in terms of rewards, what are the practical implications of under-sampling? And whether, given a sampling strategy similar to that of the Bayesian model, the subjects would indeed achieve better results or reach a plateau?

- Minor comments

- o How did the researchers correct the comparisons for multiple corrections?

- o There is a typo in line 548

- o Lines 582-583 are different from what the figure shows.

Reviewer #2 (Remarks to the Author):

This revision is a substantial revision from the previous version. The new version addresses nearly all of the issues I raised in the last round:

- providing evidence that their studies could shift sampling rate

- using statistical inference methods that allow one to collect evidence in support of the null (i.e., Bayes factors)

- using models to get at a more mechanistic explanation

The paper and research is well executed and I found the results on undersampling and explanation to be informative.

One comment that would help give me more confidence is to see model fits to the actual data. How well do the models here recreate the behavior. I may have missed this, but the modeling results are relative model comparisons. It would be nice to see how well the models recreate the data.

EDITORIAL POLICIES

We ask that you ensure your manuscript complies with our editorial policies and reporting requirements.

To that end, we require revised manuscripts to be accompanied by two completed items: a reporting summary that collects information on study design and procedure, and an editorial policy checklist that verifies compliance with all required editorial policies.

- <https://www.nature.com/documents/nr-reporting-summary.zip>>Nature Research Reporting Summary
- <https://www.nature.com/documents/nr-editorial-policy-checklist.pdf>>Editorial Policy Checklist

All points on the policy checklist must be addressed. Your revised manuscript can only be sent back to the referees if these checklists are completed and uploaded with the revision.

Notes: If you have submitted a Stage 1 Registered Report, Review, Primer, Comment, or Perspective you do not need to submit these forms. If you have already submitted these forms, you may disregard this request.

** Visit Nature Research's author and referees' website at <http://www.nature.com/authors>>www.nature.com/authors for information about policies, services and author benefits**

Communications Psychology is committed to improving transparency in authorship. As part of our efforts in this direction, we are now requesting that all authors identified as 'corresponding author' create and link their Open Researcher and Contributor Identifier (ORCID) with their account on the Manuscript Tracking System prior to acceptance. ORCID helps the scientific community achieve unambiguous attribution of all scholarly contributions. You can create and link your ORCID from the home page of the Manuscript Tracking System by clicking on 'Modify my Springer Nature account' and following the instructions in the link below. Please also inform all co-authors that they can add their ORCID to their accounts and that they must do so prior to acceptance.

If you experience problems in linking your ORCID, please contact the <http://platformsupport.nature.com/>>Platform Support Helpdesk.

Version 2:

Decision Letter:

Dear Nick,

Your manuscript titled "Biased expectations about future choice options predict sequential economic decisions" has now been seen by reviewer #1, whose comments appear below. In light of their advice I am delighted to say that we are happy, in principle, to publish a suitably revised version in Communications Psychology.

We therefore invite you to revise your paper one last time to address the remaining concerns of our reviewers and a list of editorial requests. At the same time we ask that you edit your manuscript to comply with our format requirements and to maximise the accessibility and therefore the impact of your work.

EDITORIAL REQUESTS:

Please review our specific editorial comments and requests regarding your manuscript in the attached "Editorial Requests Table". Please outline your response to each request in the right hand column. Please upload the completed table with your

manuscript files as a Related Manuscript file.

SUBMISSION INFORMATION:

OPEN ACCESS:

* DATA AVAILABILITY:

Link Redacted

Best regards,

Marika

Marika Schiffer, PhD
Chief Editor
Communications Psychology

REVIEWERS' EXPERTISE:

REVIEWERS' COMMENTS:

Reviewer #1 (Remarks to the Author):

The authors have done a great job addressing all of my comments. I have no further ones.

Reviewer #1 (Remarks to the Author):

In this manuscript the authors examined sampling behavior in full information stopping problems. First, in a pilot study, they showed that the participants' sampling rate is lower (numerically) than that of a Bayesian ideal observer model that relied on the objective price values (undersampling bias), but higher than that of an ideal observer model that relied on the subjective values (oversampling bias). Then, in their main study, they showed that the sampling rate of the participants did not significantly change across different task features. However, the classification of the participants' performance (undersampling or oversampling) changed as a result of the model it was compared to: a comparison to model 1 (objective values) resulted in undersampling bias (or no bias at all), while a comparison to model 2 (subjective values) flipped the results. The authors suggest that this pattern of results stems from differences in the prior distributions of the objective (model 1) and the subjective values (model 2).

The results are novel and interesting and the manuscript is well-written and easy to follow. I have a few suggestions that I hope would help to improve it.

We appreciate the reviewer's positive response and constructive attitude towards improving the manuscript. Please note that the narrative has developed further, compared to this summary, as a result of the new analyses and empirical data, which have been introduced in the response to the reviewers' and editor's comments.

1. Payoff schemes – I found the different payoff schemes (Reward 1 & Reward) a bit problematic. The authors mention that “payoff schemes at their most potent cannot switch between under- versus over-sampling biases” (Supplementary Text C). However, the different payoff schemes can almost completely eliminate the bias. That is, to change the classification of behavior from under/oversampling bias to no bias at all.

It is true that incentivisation scheme may not be able to (by itself) completely change the bias from undersampling to oversampling, using our previously-submitted paradigms. Nevertheless, part of our main narrative indeed claims that participants generally are reluctant to increase their sampling rates. Thus, in the new manuscript, we move this incentivisation issue to the forefront (as, we suspect, the reviewer is suggesting we should do) and we now draw the interpretation that this is an example where it is optimal under some incentivisation schemes to increase the sampling rate and participants do not do this. This interpretation is consistent with the reviewers (correct) observation that “the different payoff schemes can almost completely eliminate the bias. That is, to change the classification of behavior from under/oversampling bias to no bias at all”. Based on our new computational models, participants may be reluctant to increase their sampling rates because of a pessimistic prior expectation about upcoming samples.

2. Statistical backup – In several places the authors interpret the results based on visual inspection without backing up their claims with statistical analyses. For example, in Figure 4A the subjective values are presented as a function of the objective prices, and the relationship between them is described as ‘sigmoidal’. However, this relationship should be examined more quantitatively. For instance, the authors could fit a mixed model sigmoidal function to the data and compare its fit to other models (e.g., linear model, decaying exponential function, etc.). In addition, the authors mention that the prior distributions in Figure 4B are different, but do not show a significant difference between them (e.g., using a Kolmogorov–Smirnov test). Finally, in Supplementary Text C, the authors compare different conditions without performing any statistical analyses (for example, is the mean sampling rate in Model 1 – reward 1 significantly higher than the mean sampling rate in Model 1 – reward 2? etc.).

In light of the changed results, these comparisons of subjective versus objective values are now moot and so no longer reported in our manuscript. Our newly analysed results (e.g., the newly-added Studies 2 and 3) suggest that the use of subjective and objective values in the ideal observer model produces relative small and inconsistent differences, compared to factors like the payoff scheme or sequence length.

3. Additional Analyses/discussions – In my opinion, several statistical analyses/discussions could be added to better understand the data and strengthen the manuscript: Learning effects – Did the participants show any learning effects during the studies? For example, did they move from undersampling to oversampling across trials (or vice versa)?

As with all of our studies using full information problem paradigms, and as has been previously reported in the literature (Lee, 2006), there are no compelling learning effects to report in our data. We have added some content to the Discussion stating this.

We show below plots of participants’ sampling rates in each of our studies as a function of sequence number (As the ideal observer treats each sequence independently, there can be no ideal observer differences in sampling rate). In some of these plots, there appears to be a hint of a slight decrease in sampling rate, but any such effect appears small (mainly within the error bars, which are 95% confidence intervals of each mean) and do not replicate well over studies. We feel it would be a risk to attempt to draw an inference on the basis of these results.

While learning effects are certainly an interesting research question in their own right (and presumably the reason they have already received attention in the literature before this), it is not clear to what extent learning effects would directly support or deny our main conclusions.

Pilot baseline and Pilot full

Study 1 baseline, squares, timing and payoff

Study 1 ratings and full

Study 2

Study 3, sequence length 10 and sequence length 14

Bayesian models – Models 1 & 2 provide benchmarks to which participants' sampling rate is compared. Could the authors add a discussion (or analysis) about whether these models can also account for the cognitive mechanisms underlying the behavior of the participants.

The largest change to the new manuscript is the introduction of these theoretical computational models, as requested. In response to this request by both reviewers and the editor, we have included some of our new work, which was indeed designed exactly to “account for the cognitive mechanisms underlying the behavior of the participants”. We show that suboptimal decisions on this task are theoretically accounted for by biased expectations of future option values.

·Sampling efficiency – It would be interesting to compare the mean payoff obtained by the participants to that obtained by the optimal models.

In the Supplementary Materials, we have added plots of the mean rank of the chosen prices for each condition / study. This new analysis adds an important

dimension to our narrative, as it can be seen that participants perform close to optimally, despite sampling less than optimally.

· Rating consistency – Each price was presented twice at the rating phase. What was the correlation between the two ratings?

The correlations (now reported in the respective Methods sections) are Pilot full: 0.83, Study 1 full: 0.87, Study 1 ratings: 0.81, Study 2: 0.85, Study 3 10 options: 0.88, Study 3 14 options: 0.84.

· Subjective vs. Objective values – The authors speculate that participants sampling rate would be affected by the distribution of subjective perception of prices (e.g., £550 is roughly equal to £400, p. 10/ first paragraph). This can be empirically examined by comparing participants the sampling rate of participants with a relatively linear relation between the subjective values and objective prices, to participants with a more ‘curvy’ relation.

As we have changed our claims with respect to subjective versus objective values, this comparison is now moot.

· Effect Size – The authors reported only significance levels, but not effect sizes.

We report effect sizes for all comparisons involving human behaviour in the new Figure S4 in the Supplementary materials that visualises the effect sizes for all pairwise comparisons in Study 1 (i.e., the null effects, in which participants appear to sample at nearly the same rate over conditions).

4. Code availability – As this manuscript compares human sampling behavior to that of ideal observer models, it would be great if the authors could make the code used to implement the ideal observers publically available.

https://github.com/nicholasfurl/Model_fitting_hybrid_study

Reviewer #2 (Remarks to the Author):

Summary

The paper investigates full information optimal stopping problems, specifically when people oversample/undersample in this scenario. The author first hypothesized that

the number-based tasks led to undersampling and picture-based tasks led to oversampling (as reported by previous studies). But oversampling was observed in the number-based task in the Pilot study, which indicates that pictorial stimuli may not be the only reason causing oversampling. The goal of the Main study is to isolate which task feature leads to oversampling in number-based tasks. They found the human sampling rate is unchanged in all six conditions (i.e., Baseline, Full, Squares, Payoff, Timing, and Ratings). The conclusion about over versus undersampling is determined by the implementation of the model. The paper implemented two models with different prior generating distributions for the Bayesian optimality model. Model 1 uses objective prices as the prior generating distribution, and Model 2 uses subjective evaluations of prices.

This accurate summary of our previous submission no longer holds in its entirety for the new submission. Our two new studies, reanalysis of our existing data and the addition of our theoretical model building and comparison has shifted our narrative and conclusions.

We still contend that participants adopt a relatively rigid sampling strategy. Though we show that manipulations within our paradigm can shift the sampling rates of the ideal observer and of the participants, the former is affected more easily than the latter.

As requested by this reviewer, we show that manipulations of our paradigm can in fact shift participants' sampling rate to some degree. In the new Study 3, we replicate a previous finding using a version of our paradigm (Costa & Averbeck, 2015) that participants sample more for longer sequences. But in that previous study, and in our replication of it, the ideal observer increased its sampling rates numerically more than participants, leading to greater undersampling at longer sequence lengths.

Moreover, the ideal observer samples more when the incentivisation schemes rewards only the top three ranks than the full conditions, where the incentivisation schemes reward any choice (proportional to its option value). Although our paradigm is capable of shifting the ideal observer's sampling rate and we can statistically detect this shift, these are not the case for participants. Undersampling is greater in conditions where the incentivisation schemes rewards the top three ranked choices, because the ideal observer increases its sampling rates and participants not so much.

As in the previously submitted manuscript, a number of other methods features failed to affect the sampling rates for either the ideal observer or for the participants. We now report Bayesian tests, which have sufficient sensitivity to show statistical evidence favouring the null model of equal means over models where means differ.

Following this reviewer's advice, we now provide theoretical models to explain why participants do not raise their sampling rates as much as we might expect them to. Models that undersample because they adopt too pessimistic a prior distribution over predicted option values best fit participants' sampling behaviour, compared to a number of competing models of suboptimal choices on our task.

Review

The potential contributions of the paper are to show that some task features are insignificant in affecting human sampling biases and that the conclusions about under/oversampling are completely model-based. However, the paper falls short in several places. It is not clear that the study itself was sensitive enough to detect any changes in behavior so it gives low confidence in the conclusion that human sampling is unaffected by task features.

We have provided the requested new study to show successful experimental manipulation of participants' sampling rates.

We now report Bayesian tests, which are sufficiently sensitive to show positive statistical evidence for equivalent means between the various methods conditions in question.

We further note that our design was sensitive enough to detect changes in ideal observer behaviour (at least for the different payoff schemes), even when it finds no differences in participants' sampling rates.

The paper does not give any insight as to why there is a difference between the Bayesian optimal models and human behavior; the only conclusion here is that participants under sample the Bayesian optimal model, but even that conclusion, as the paper establishes, is completely model-based.

Our revised manuscript now gives "insight as to why there is a difference between the Bayesian optimal models and human behavior". We have built a number of models that could explain undersampling and, in the new manuscript, we fit them to participants' decisions. The ensuing model comparison replicates highly across the studies and conditions in suggesting that participants do not increase their sampling rates in the same conditions as the ideal observer does because they subjectively expect future samples to be lower valued than they should.

Finally, overall the paper was very difficult to parse.

We are submitting effectively an entirely new paper. Nevertheless, we are of course happy to entertain any specific or constructive suggestions for rewriting.

Comments

1. ****Sensitivity to task features.**** The author state, "Our Pilot Study shows that

oversampling is not limited to tasks that present option values as pictures but can also occur for some tasks using numeric stimuli to communicate option value. Our Main Study then attempted to systematically isolate which task feature leads to oversampling on number-based tasks.” The idea here was that there were several features in the experimental protocol that apparently differed between the current numeric version and the past numeric versions that established undersampling. The Main study then changed these features to see if that could explain the difference between the current result and the past result. But, one reasonable issue here is a sensitivity issue. Could it be that in this particular study, participants are just relatively insensitive to these task features? That is, they should show that the study could actually prompt changes in behavior and that the study can detect it. That would give confidence in the conclusion that participants are insensitive to task changes.

Costa & Averbek (2015) using the methods we adapted for our study, already showed that sequence length increases participants' sampling rates. We now replicate this finding in Study 3. Therefore, some manipulations using this paradigm can indeed affect participants' behaviour. We have pre-registered and *a priori* powered Study 3, as requested by the editor.

Our Study 2 is sufficiently sensitive to detect positive evidence for null models where means are equal, using Bayes Factors (Figure 2).

We note that our paradigm methods did suffice to modulate the ideal observer's sampling rate in response to payoff scheme and sequence length, even if participants showed no modulation (payoff scheme) or less modulation (sequence length), suggesting the design elements capable of modulating sampling for an ideal observer are nevertheless present.

We note that no evidence has been (explicitly) mentioned that confirms any specific problem with our paradigm that would prevent detection of participant shifts in sampling.

2. *****Model 1 and Model 2***** *A real struggle in this paper is the use of Model 1 and Model 2 and the definition of undersampling and oversampling. Over many, many reads. Here is what I understand. Model 1 is the Bayesian Optimal model with objective values. Model 2 is the Bayesian Optimal Model with subjective values. Many papers showed in these optimal search tasks that people under sample with references to the Bayesian Optimal Model with objective values (Model 1). A couple of papers (Furl et al., 2019; van de Wouw et al., 2022) came out using images instead of numeric values, which meant researchers needed to collect subjective values of the images to run the Bayesian Optimal Model. But, comparing behavior to this model led to the conclusion that people oversample. This paper set up a situation where both Model 1 and Model 2 could be used, establishing that the conclusion of undersampling (with reference to Model 1) and oversampling (with reference to Model 2) is completely model-based. This is useful information, but honestly, it takes a lot of effort for the reader to figure this conclusion out. The writing obscures this because it flips between Model 1 and Model 2. But, there could be a lot more work done to make that clear. For instance, in the Pilot study, it is really*

confusing to read that “Our Pilot Study shows that oversampling is not limited to tasks that present option values as pictures but can also occur for some tasks using numeric stimuli to communicate option value. Our Main Study then attempted to systematically isolate which task feature leads to oversampling on number-based tasks.” But, as a reader, you are looking at Figure 1 and Model 1 and saying there is no oversampling.

We suspect that the new manuscript may not pose the same challenges, as we have abandoned the Model 1 / Model 2 notation and, more generally, the new narrative is no longer rooted in comparisons of these two model types.

A deeper issue here is that, in the end, the main conclusion of the paper is that the conclusion of over vs undersampling is completely model-based. But, this is not that interesting because it is true by definition.

Given the conclusions drawn from the new empirical data and theoretical models, this comment is now moot.

There is really little to no insight as to why human behavior differs from the optimal math solution. The paper's opening suggests it is going to investigate this, but the studies do not do this. The main study seems to have been designed to do this, but really all we learn is that changes in the experimental task features did not have any impact. And as mentioned above, there is a real issue here that it could be that the experiment itself just can't show or detect any changes in human behavior.

The new manuscript is now centred on theoretical model fits and comparisons that give insight into “why human behavior differs from the optimal math solution”.

The paper could also generate models that closely replicate human stopping behavior and find the best model by comparing the models' results with human behavior. The authors seemed to try to do this with, for instance, the subjective value version of the Bayesian model. But, then, the experiments give us no information about why there are differences between human behavior and the models.

We have followed this reviewer's advice and, indeed, our new model comparison has highlighted a possible computational mechanism for explaining undersampling bias, which replicates well across a number of studies.

As this reviewer prefers, the new manuscript is now centred on theoretical model fits and comparisons that give insight into “why there are differences between human behavior and the models”.

Moreover, it isn't really clear if the Bayesian optimal models would have predicted any changes based on the manipulations in the main study.

The ideal observer sampling rates for the different manipulations in the main study (now labelled Study 1 in the new manuscript) are visible in Figures 4 and 5.

Minor Comments

1. *The difference between 1 and 10 is not the same as between 1000 and 1010.*

This sentence is no longer in the manuscript.

2. *"Although full information problems lack the many restrictive assumptions of the secretary problem, they cannot employ such a simple rule to derive optimal performance for comparison with human performance. "*

3. *The paper states that secretary problem is a simpler problem of full information problems, and the simple mathematical rule of the secretary problem cannot be used to derive optimal performance for comparison with human performance. But the secretary problem is known as the "no-information game", the assumptions of the no information game are very different from those of the full information game. Thus, the optimal solution of no information game is of course not applicable to the full information game.*

The reviewer appears to agree with us that optimal solution to the Secretary problem does not apply to full information problems. To avoid any misunderstanding, the revised manuscript provides more thorough comparison of these different decision problems in the Methods, when we introduce the cut off (CO) model, which is derived from the solution to the secretary problem. We also give more text over to a critical discussion of the merits of Peter Todd's and Geoffrey Miller's (1999) claim that the CO model should be applied beyond the secretary problem.

Reviewer #1 (Remarks to the Author):

I thank the authors for addressing my previous comments and for the considerable effort invested in revising the manuscript. In this revised version of the manuscript, the authors have added several theoretical models aimed at explaining the behavior of the participants. Generally, the addition of these models improved the manuscript. However, since the article has undergone considerable changes, I have a few additional comments regarding the new version of the manuscript, both in relation to its clarity and in relation to the newly added models.

- *Clarity and presentation - The introduction and paradigm description remain relatively clear. However, the model description and results sections are relatively complex and less intuitive. I suggest several improvements:*

We appreciate the reviewer's appreciation of the new modelling contributions! And we of course are pleased to implement these constructive comments to help improve clarity.

o Ideal observer model - The basic model is explained in a relatively abstract way. A simple numerical example or visualization (e.g., one that illustrates the backwards induction) could help readers who are not familiar with this model to understand it better.

We have added a section to the supplementary materials that takes the reader through an intuitive and more concrete step by step description of backwards induction.

o Abbreviations - Throughout the article, the authors used abbreviations such as CO, CS, etc. Using the full names of the models could enhance readability and flow of the text.

We now spell out these abbreviations.

o A table summarizing the key features of each model could help.

We have created a new Table 1 for this purpose.

o Figure 1 - The font size is relatively small, making it difficult to see what is written.

As these are screen shots of the actual paradigms, we are not able to change the font sizes without misrepresenting how the paradigm was presented. Therefore, we have reorganised the figure to enlarge the screen images to make the text more visible. This means individual paradigms cannot be separate panels. So we have created multiple figures, one for each paradigm, enlarged to span each page, in the Supplementary Materials. We have kept the baseline paradigm (previously panel A) as Figure 1 in the main text to keep one illustration of how a paradigm of this kind generally appears.

o Figures 2-7 - i. Adding a legend explaining the color coding would assist in understanding the Figure more easily. ii. Some of the BIC values look almost identical, and it is very difficult to distinguish between them. Adding numerical BIC values would facilitate easier comparisons between the models. iii. Figure 3 - Column headings would make the figure clearer.

We have added all these features to the figures.

- *Qualitative versus quantitative comparison - While the BP model's superior BIC values indicate a better fit, it would be helpful to understand the specific qualitative aspects of the data that contribute to this superiority. For example, how does the BP model's prediction of participant behavior differ from the CS model (i.e., both models predict under-sampling, so where are their predictions different)? Providing a more detailed comparison of the predictions and behavior patterns could illuminate why the BP model outperforms others.*

The top rows of the revised Figures 3-7 report the mean sampling rates and in the top rows of Figures S7, S9-S11 and S14 the mean ranks of chosen options for all the models. We plot these measures alongside participant behaviour for direct comparison of whether the models reproduce participants' performance. It can be seen that all models reasonably reproduce participants' sampling rates. It can also be seen that, at the level of mean ranks of chosen prices, the models do diverge. Specifically, the Cut Off heuristic fails to reproduce participants' performance while the Cost to Sample and Biased Prior models can.

We have added two new analyses, exploiting the large sample size in Study 2. In Figure S12, we report relationships between individual differences in participants' versus models' sampling rates. This analysis responds to the reviewer's request to report "*how does the BP model's prediction of participant behavior differ from the CS model*". We find that both Biased Prior and Cost to Sample models show robust correlations. In Figure S13, we also provide an analysis of participant and model choice thresholds. The reviewer has not specified what other "detailed comparisons" would be desired but we would be happy to consider other analyses.

We have also added to the General Discussion the important point that all the models appear to well predict participant behaviour and, in some cases, the Cost to Sample model also fits a remarkable portion of our samples (e.g., in Study 3). We suspect that where model predictions differ might be subtle and vary from participant to participant, and we recommend using paradigms that intentionally manipulate factors that should dissociate the models.

In our revised General Discussion beginning line 700 we address these issues:

"We should note, however, that the Cost to Sample model – in which participants perceive sampling itself to be costly or rewarding – was the best-fitting model for a substantial number of participants and therefore may explain suboptimal decisions in a subset of our participants. Indeed, all our models accurately predicted participants' mean sampling rates (Figures 3-7). The models did however diverge in their predictions of participants' mean rank of chosen options (Figures S7, S9-S11, S14). Here, the Cut Off heuristic was unable to obtain similar levels as participants, while Biased Prior and Cost to Sample models could. Our supplementary analyses of the large sample in Study 2 show individual differences in participant sampling rates were highly correlated with sampling rates predicted by all three models (Figure S12). And, participants' sequence specific thresholds were approximated by all three models (Figure S11). The framework we promote here, therefore – using an ideal observer to model accurate performance and then parameterising it to account for systematic bias – appears to produce models that predict participant data with reasonable accuracy. Moreover, different participants in the same sample might adopt any of these strategies, even if the Biased Prior strategy might be the most common. In some cases (as in Study 3), the Cost to Sample model best fitted a remarkable share of individual participants. Given the high predictivity of all our models, it can be difficult to discern exactly what choices the Biased Prior is superior at predicting, compared to other models. Our recommendation is that these models be compared on paradigms specifically designed to test this hypothesis. For example, manipulations of participants' expectations about upcoming option values (i.e., their prior) should produce the types of systematically different decisions that would be predictable from a Biased Prior model."

- *Additional model considerations - The models implemented provide valuable insights, but exploring additional models could yield even richer findings.*

- o For example, did the authors examine a combined BP and CS model? Since the mechanisms underlying these two models are different, a combined model might be able to achieve better results.*
- o Another possible model is one similar to BP/CS but assumes time-varying parameters (for example, that the sampling cost increases over time). This idea is similar to the urgency signal/collapsing boundaries in sequential sampling models.*
- o The CS and BP models are essentially versions of the IO model. There may be models from different families that might explain the participants' undersampling. For example, a model that assumes that the participants scanned the space of possibilities partially (e.g., only for 3-4 levels), and therefore undersampled.*
- o Exploring some of these models (or at least discussing them as potential avenues in the discussion section) could strengthen the manuscript.*

We agree with the reviewer that the introduction of the modelling framework in this paper opens many new avenues for theory development and we share enthusiasm for these possibilities. We have added text in our General Discussion outlining these interesting new research directions, as advised by the reviewer:

“We have introduced a framework whereby optimality solutions including that of the Secretary Problem and that of the full information problem have been leveraged to explain accurate performance on optimal stopping tasks. And our framework has taken the approach of parameterising these models to explain systematic sources of suboptimal performance. Given that we have proposed this framework and demonstrated its utility, we expect that future research can refine the models we have proposed or build improved models that may better fit participant data. For example, more complex models that combine multiple bias-related free parameters (e.g., the cost to sample parameter and a constant added to the prior mean as fitted parameters at the same time) might be considered. Also, more sophisticated versions of our models might be formulated, such as cost to samples that change across sequence position Future research might explore models that use a limited-capacity backwards induction, which can only partially explore possible future states, or use a simpler heuristic to approximate the choice threshold / value of sampling again.”

- *Model feasibility - In the context of the previous comment, the BP model outperformed the other models in terms of BIC results. However, the feasibility of the calculation required by it raises questions about its psychological plausibility. How realistic is it for participants to perform such complex calculations? Could simpler heuristics underlie what appears to be complex computation?*

We have added to the General Discussion some text addressing this issue, including citation of previous evidence from fMRI that neural responses may encode quantities computed by these kinds of models. We note also that we have already included in our model comparison a simpler heuristic (the Cut Off heuristic) and the model fitting evidence appears to be against this heuristic and more in favour of the models based on the ideal observer.

We write in the General Discussion starting line 799:

“When considering models that might be built and tested in the future, it is worth considering that the Bayesian ideal observer solution to the full information problem (which we used as a base for some of our models) is relatively computationally complex, especially its backwards induction algorithm (See Supplementary Materials for more information). Future research might explore models that use a limited-capacity backwards induction, which can only partially explore possible future states, or create some simpler heuristic to approximate the choice threshold / value of sampling again. We note that already our evidence here undermines the case for a previously proposed heuristic, the Cut Off heuristic. In any case, we do not know the capacity of the neural architectures involved in solving these problems, rendering it difficult to reject models a priori on this basis. Indeed, it is plausible that neurons may be implementing similar computations as the kinds of models we investigated here. It has already been shown that brain responses correlate trial-by-trial with fluctuations in quantities derived from backwards induction based optimal stopping models that have

been fitted to human participant choice data (Costa & Averbeck, 2016; Furl & Averbeck, 2011).”

- *Optimality - Given that human performance closely matched the optimal model in term of rewards, what are the practical implications of under-sampling? And whether, given a sampling strategy similar to that of the Bayesian model, the subjects would indeed achieve better results or reach a plateau?*

We have expanded on our existing text in the General Discussion regarding the comparison of ranks of chosen options between participants and the Ideal Observer.

“We have already mentioned several ways that the modelling framework we propose here and the results we have obtained raise new research questions and open new research lines. One last issue that we also feel deserves further study relates to the extent to which systematic bias translates into real losses for people confronted with real optimal stopping problems. We have already proposed above an interesting theoretical possibility that biases like Biased Prior strategies might have an adaptive function, so long as they can maintain near-optimal performance. Indeed, within the narrow range of sequence lengths and domains (i.e., smartphone prices) that we have examined here, participants’ biased choices largely satisfied, and produced performance that, while not optimal, did not cause a striking loss for participants, when measured as rank of chosen option. Nevertheless, we also show herein that factors such as sequence length and incentivisation can affect the size of bias (e.g., longer sequences increase undersampling bias, as the Ideal Observer adjusts to the longer sequences but participants less so). Just how large biases can eventually become and the extent to which significant losses might accrue for agents due to ever larger biases cannot be answered directly by our data and would benefit from more targeted investigations. Ideally, future studies could better approximate real-world conditions of decision making, or even collect field data “from the wild”, to assess what kinds of losses might (or might not) occur under more ecologically valid conditions, as opposed to the more tightly controlled studies we report here.”

- *Minor comments*

- o How did the researchers correct the comparisons for multiple corrections?*

We have clarified in the text that the corrections were Bonferroni.

- o There is a typo in line 548*

We have corrected this.

- o Lines 582-583 are different from what the figure shows.*

Thank you for identifying this typo. We have corrected “undersampling” to “oversampling”.

Reviewer #2 (Remarks to the Author):

This revision is a substantial revision from the previous version. The new version addresses nearly all of the issues I raised in the last round:

- *providing evidence that their studies could shift sampling rate*
- *using statistical inference methods that allow one to collect evidence in support of the null (i.e., Bayes factors)*
- *using models to get at a more mechanistic explanation*

The paper and research is well executed and I found the results on undersampling and explanation to be informative.

We are pleased the revision addresses this reviewer's concerns and for this positive response.

One comment that would help give me more confidence is to see model fits to the actual data. How well do the models here recreate the behavior. I may have missed this, but the modeling results are relative model comparisons. It would be nice to see how well the models recreate the data.

The upper panels of figures 3-7 show the sampling rates of the models and how they compare (favourably) to participants' mean sampling rates, suggesting that they do reproduce the results. The Supplementary figures show the models' mean ranks of chosen options alongside those of participants. With the exception of the Cut Off heuristic (which consistently and considerably undershoots participants' performance), the other models perform similarly to participants.

We have now added scatterplots and correlation values (Figure S12), showing that sampling rates are also well-predicted by all models at the level of individual participants, using the relatively large sample size in Study 2. Also from Study 2, we report a new analysis of thresholds at different sequence positions in Figure S13.

We hope these collectively demonstrate the value of the Biased Prior model for predicting sampling choices in the majority of participants.